



# Impact of mid-glacial ice sheets on deep ocean circulation and global climate: Role of surface cooling on the AMOC

Sam Sherriff-Tadano[1], Ayako Abe-Ouchi[1], Akira Oka[1]

[1]Atmosphere and Ocean Research Institute, the University of Tokyo, Kashiwa, Japan

*Correspondence to*: Sam Sherriff-Tadano (tadano@aori.u-tokyo.ac.jp)

**Abstract.** This study explores the effect of southward expansion of mid-glacial ice sheets on the global climate and the Atlantic meridional overturning circulation (AMOC), as well as the processes by which the ice sheets modify the AMOC. For this purpose, simulations of Marine Isotope Stage (MIS) 3 and 5a are performed with an atmosphere-ocean general circulation model. In the MIS3 and MIS5a simulations, the global average temperature decreases by 5.0 °C and 2.2 °C, respectively, compared with the preindustrial climate simulation. The AMOC weakens by 3% in MIS3, whereas it is enhanced by 16% in MIS5a, both of which are consistent with a reconstruction. Sensitivity experiments extracting the effect of the expansion of glacial ice sheets from MIS5a to MIS3 show a global cooling of 1.1 °C, contributing to about 40% of the total surface cooling from MIS5a to MIS3. These experiments also demonstrate that the ice sheet expansion leads to a surface cooling of 2 °C over the Southern Ocean as a result of colder North Atlantic deep water. We find that the southward expansion of the mid-glacial ice sheet exerts a small impact on the AMOC. Partially coupled experiments reveal that the global surface cooling by the glacial ice sheet tends to reduce the AMOC by increasing the sea ice at both poles, and hence compensates for the strengthening effect of the enhanced surface wind over the North Atlantic. Our results show that the total effect of glacial ice sheets on the AMOC is determined by the two competing effects, surface wind and surface cooling. The relative strength of surface wind and surface cooling depends on the ice sheet configuration, and the strength of the surface cooling can be comparable to that of surface wind when changes in the extent of ice sheet are prominent.

## 1 Introduction

During the last glacial period, ice sheets evolved drastically over the Northern continent (Lisiecki and Raymo 2005, Clark et al. 2009, Grant et al. 2012, Spratt and Lisiecki 2016). After the initiation of the northern glacial ice sheets at the end of the Last Interglacial, the ice sheets expanded over northern North America and northern Europe during the early glacial period, Marine Isotope Stage (MIS) 5d-a, and further expanded during MIS4 associated with weakening of summer insolation. Then, the glacial ice sheets once shrank during the mid-glacial period (MIS3), when the summer insolation and the concentration of $CO_2$ were relatively large (Abe-Ouchi et al. 2007, Grant et al. 2012, Spratt and Lisiecki 2016, Pico et al.



2017). Subsequently, the ice sheets further expanded during MIS2, when the summer insolation and the concentration of CO$_2$ were low, and reached their maximum volume at the Last Glacial Maximum (LGM, Peltier 2004, Clark et al. 2009, Tarasov et al. 2012, Ishiwa et al. 2016). Because of these drastic differences in the ice sheet and climate compared with modern times, the last glacial period is considered as important to improve the understanding of the effect of ice sheets on climate.


Previous studies investigated the impact of glacial ice sheets on the climate under the LGM, which is set as the target period in the Paleoclimate Model Intercomparison Project (PMIP, Braconnot et al. 2007, Braconnot et al. 2012, Abe-Ouchi et al. 2015, Kageyama et al. 2017). Based on reconstructions, the climate of the LGM is known to be the coldest and most stable period of the last glacial (Kindler et al. 2014, Kawamura et al. 2017). Furthermore, the Atlantic meridional overturning

circulation (AMOC) is considered to have been shallower and perhaps weaker compared with the preindustrial era (McManus et al. 2004, Bohm et al. 2015, Muglia et al. 2018, Menviel et al. 2020). Modelling studies show that the expansion of the glacial ice sheet cause a large cooling, a strengthening of atmospheric circulation, and a southward shift of the rain belt over the North Atlantic (Cook and Held 1988, Kageyama et al. 2000, Abe-Ouchi et al. 2007, Laine et al. 2009, Pausata et al. 2011, Hofer et al. 2012, Lofverstorm et al. 2014, Merz et al. 2015). These studies also show that the response

of the atmospheric circulation is largely affected by the height of the ice sheet (Gong et al. 2015, Merz et al. 2015), while the strength of the surface cooling is largely affected by the extent of the ice sheet (Abe-Ouchi et al. 2007).

Several studies using an atmosphere ocean coupled general circulation model (AOGCM) also show that the glacial ice sheets exert a large impact on the AMOC. Many of these studies show a strengthening of the AMOC in response to the expansion

of the northern glacial ice sheet (Eisenman et al. 2009, Brady et al. 2013, Zhang et al. 2014a, Gong et al. 2015, Klockmann et al. 2016, Brown and Galbraith 2016, Kawamura et al. 2017), while one study shows a reduction of the AMOC (Kim 2004). From sensitivity experiments, it is shown clearly that the higher glacial ice sheets enhance the surface wind as well as the wind-driven oceanic transport of salt into the deep-water formation region over the North Atlantic, which increases the surface salinity and causes a strengthening of the AMOC (Oka et al. 2012, Muglia and Schmittner 2015, Sherriff-Tadano et

al. 2018). Other studies also suggest the importance of changes in surface cooling (Smith and Gregory 2012), which can cause either a strengthening or weakening of the AMOC by enhancing deep-water formation over the North Atlantic (Schmittner et al. 2002, Oka et al. 2012, Smith and Gregory 2012) or by increasing the amount of sea ice over the northern North Atlantic and Southern Ocean (Kawamura et al. 2017). Nevertheless, due to the complicated coupling between the atmosphere and ocean in climate systems, the role of surface cooling by the glacial ice sheets on the AMOC still remains

elusive.

While the effects of glacial ice sheets on the LGM climate gain large attention, the effect of pre-LGM glacial ice sheet on the global climate and AMOC is less explored. Reconstructions of the ice sheets prior to the LGM still have large uncertainties,





though recent studies suggest some differences in ice sheets between the early glacial (MIS5a) and mid-glacial (MIS3).
Between these two periods, the volume of ice sheets is slightly larger in MIS3 than in MIS5a (Lisiecki and Raymo 2005, Grant et al. 2012, Abe-Ouchi et al. 2013, Spratt and Lisiecki 2016, Pico et al. 2017, Willeit and Ganopolski 2018). In addition, studies with ice sheet modelling suggest a larger extent of the North American ice sheet in MIS3 compared with MIS5a, despite small differences in the maximum height of the ice sheet (Fig. 1a, Abe-Ouchi et al. 2007, 2013, Niu et al. 2019). This is different from what is revealed with explorations using the LGM ice sheet, whose changes are large in both
the height and extent. Hence, by comparing the early-glacial and mid-glacial ice sheets, one may obtain different responses in the AMOC and global climate, whose effect of surface cooling is prominent.

Furthermore, recent reconstructions show some discrepancies between the MIS3 and MIS5a climates. For example, it is shown that the AMOC is slightly weaker in MIS3 compared with that of MIS5a (Bohm et al. 2015). Ice core data also show
that the duration of the millennial time-scale climate variability is shorter in MIS3 compared with MIS5 (Capron et al. 2010, Buizert and Shcmittner 2015, Lohmann and Ditlevsen 2019). Hence, by exploring the impact of the mid-glacial ice sheets on the global climate and AMOC, one can also assess the potential role of differences in ice sheets between MIS3 and MIS5a in causing the differences in climate and AMOC between MIS3 and MIS5a.

In this study, we investigate the impact of the expansion of the mid-glacial ice sheets on the global climate and the AMOC. Specifically, we explore how the differences in the ice sheets between MIS3 and MIS5a have an impact on global climate and AMOC. For this purpose, we perform climate simulations of MIS3 and MIS5a with a comprehensive climate model. Furthermore, we explore the processes by which the changes in the ice sheet modify the AMOC. Particularly, we focus on the role of changes in surface cooling by the glacial ice sheets on the AMOC, which is also considered important in driving
the AMOC changes (Loving and Vallis 2005, Arzel et al. 2010, Oka et al. 2012, Sun et al. 2016, Jansen 2017), but still remains elusive in previous LGM studies. For this purpose, partially coupled experiments are conducted (Mikolajewicz and Voss 2000, Schmittner et al. 2002, Gregory et al. 2005, Sherriff-Tadano and Abe-Ouchi 2020). In this experiment, the atmospheric forcing that drives the oceanic component is switched one by one to a different forcing. For example, Gregory et al. (2005) apply this method to interpret the cause of the weakening of the AMOC in the $CO_2$ doubling simulations in the
CMIP3 models. They find that the changes in the surface heating play a large role in causing the weakening of the AMOC through reducing the heat exchange between the atmosphere and the ocean. Hence, the use of partially coupled experiments enables us to extract the effect of changes in surface cooling by the mid-glacial ice sheets on the AMOC.

This study is organized as follows. In section 2, we describe the model and the experimental design. In sections 3 and 4, we
show the results of MIS3 and MIS5a simulations, and then investigate the role of mid-glacial ice sheets on the global climate and the AMOC. The effect of surface cooling by the mid-glacial ice sheet is also explored by means of a partially coupled experiment. Sections 5 and 6 discuss and summarise the results, respectively.





## 2. Methodology

### 2.1 Model

We perform numerical experiments with the Model for Interdisciplinary Research on Climate 4m (MIROC4m; Hasumi and Emori 2004, Chan et al. 2011) AOGCM. This model consists of an atmospheric general circulation model (AGCM) and an oceanic general circulation model (OGCM). The AGCM solves the primitive equations on a sphere using a spectral method. The horizontal resolution of the atmospheric model is ~2.8° and there are 20 vertical layers. The AGCM is coupled to a land-surface model. The OGCM solves the primitive equation on a sphere, where the Boussinesq and hydrostatic approximations

are adopted. The horizontal resolution is ~1.4° in longitude and 0.56°–1.4° in latitude (latitudinal resolution is finer near the equator). There are 43 vertical layers. It is coupled to a dynamic-thermodynamic sea-ice model. Note the coefficient of horizontal diffusion of the isopycnal layer thickness in the OGCM is slightly increased to 700 $m^2$ $s^{-1}$ compared with the original model version (300 $m^2$ $s^{-1}$) that was submitted to PMIP2 [these two model versions are referred to as Model B and Model A, respectively, in Sherriff-Tadano and Abe-Ouchi (2020)]. The model version used in this study is used extensively

for paleoclimate (Obase and Abe-Ouchi 2019) and future climate studies (Yamamoto et al. 2015). It also reproduces the AMOC of the LGM reasonably well (Sherriff-Tadano and Abe-Ouchi 2020).

### 2.2 Model simulations

Three experiments are conducted with MIROC4m AOGCM (Table 1). The first experiment is named MIS5a, which aims at a period of approximately 80 ka. In this experiment, we apply a $CO_2$ level of 240 ppm, insolation of 80 ka, and an ice sheet

boundary configuration of 80 ka taken from an ice sheet model (see next paragraph for detailed information). The second and third experiments are performed under MIS3 boundary conditions, $CO_2$ of 200 ppm and insolation of 35 ka. In these two experiments, the configurations of the ice sheets differ (Fig. 1). In the second experiment, we apply an ice sheet of 36 ka (Fig. 1b). In the third experiment, we apply an ice sheet of 80 ka (Fig. 1a). These experiments are named MIS3 and MIS3-5aice, respectively (Table 1). Note that the Antarctic ice sheet is fixed to the modern configuration. The global sea level is

unchanged, and the land sea mask outside the northern glacial ice sheet region is same as the modern configuration (e.g., the Bering Strait remains open). For methane and other greenhouse gases, the concentration of the LGM is used (Dallenbach et al. 2000). Hence, by comparing MIS3 and MIS3-5aice, one can assess the impact of mid-glacial ice sheets on the global climate and AMOC. The difference between MIS3-5aice and MIS5a shows the effect of changes in $CO_2$ and insolation.

For the ice sheet forcing, we use the output from the Ice sheet model for Integrated Earth system Studies (IcIES, Saito and Abe-Ouchi 2005) driven with the climatic parameterization derived from MIROC (IcIES-MIROC, Abe-Ouchi et al. 2007, 2013). This model reproduces the evolution of the Northern Hemisphere ice sheet over the past 400,000 years (Abe-Ouchi et al. 2013), and it is used as a boundary condition for the simulations of the Penultimate Glacial Termination (Menviel et al. 2019). The model also reproduces the general pattern of the evolution of the global ice sheet volume (or equivalent sea level





change) over the last glacial period reasonably well (Abe-Ouchi et al. 2013, Fig. 1a), that is, larger ice sheets during MIS3 compared with MIS5a (Grant et al. 2012, Spratt and Lisiecki 2016, Pico et al. 2017). These volumes are the 40-meter sea level equivalent for MIS5a (approximately 33% of the LGM) and 96-meter sea level equivalent for MIS3 (approximately 80% of the LGM, Abe-Ouchi et al. 2013). The volume of the MIS3 ice sheet slightly exceeds the estimated range of sea level reconstructions (approximately 40- to 90-meter sea level equivalent during the mid-glacial, Grant et al. 2012, Spratt

and Lisiecki 2016, Pico et al. 2017). This is further discussed in section 4.

The three simulations are initiated from the previous LGM experiment of Kawamura et al. (2017), which has a weak and shallow AMOC. MIS5a is integrated for 2,000 years and MIS3 and MIS3-5aice are integrated for 3,000 years. After the integration, the AMOC settles into a vigorous mode (interstadial mode) in all experiments. Decreasing trends of deep ocean

temperature of the last 100 years are 0.002 °C in MIS5a, 0.011 °C in MIS3, and 0.007 °C in MIS3-5aice. Hence, these simulations have settled to quasi-equilibrium states (Zhang et al. 2013).

### 2.3 Partially coupled experiments

To assess the processes by which the mid-glacial ice sheets modify the AMOC, partially coupled experiments are conducted (Table 2). In these experiments, the atmospheric forcing – wind stress and atmospheric freshwater flux (precipitation,

evaporation, and river runoff) – that drives the ocean is replaced with a monthly climatology. Following previous studies (Schmittner et al. 2002, Gregory et al. 2005), the heat flux is unchanged in these experiments because fixing the surface heat condition has an unrealistic impact on the AMOC (Marozke 2012). Four partially coupled experiments are conducted based on the MIS3 and MIS3-5aice experiments (Table 2). These experiments are initiated from the last year (year 3000) of MIS3 or MIS3-5aice. The first experiment (PC-MIS3) is intended as a validation of the method. In this experiment, the

atmospheric forcing is replaced with the monthly climatology of the last 100 years of the same MIS3 experiment. Hence, the climatological atmospheric forcing is identical to that of the original experiment. The second experiment (PC-MIS3-5aice) is also conducted in a similar manner under MIS3-5aice, using the monthly climatology of MIS3-5aice. The third experiment is conducted under the MIS3 condition (PC-MIS3heat), where the wind forcing and atmospheric freshwater forcing are replaced with the monthly climatology of MIS3-5aice. Hence, the oceanic component of the model is forced by wind and

atmospheric freshwater fluxes of MIS3-5aice and the surface heat flux of MIS3. By comparing the results between PC-MIS3heat and PC-MIS3-5aice, one can evaluate the effect of surface cooling on the oceanic circulation (Gregory et al. 2005). Note that the effect of surface cooling includes changes in freshwater flux from the sea ice in addition to changes in an atmosphere-ocean heat exchange. In the fourth experiment (PC-MIS3heatano), the effect of surface cooling is evaluated in a slightly different way. In this experiment, we apply the anomalies of monthly climatology of surface wind and

atmospheric freshwater flux between MIS3 and MIS3-5aice to MIS3. Hence, this experiment is also forced by winds and atmospheric freshwater fluxes of MIS3-5aice and the surface heat flux of MIS3. By comparing the MIS3-5aice experiment with PC-MIS3heatano, we can estimate the effect of surface cooling on the oceanic circulation. The advantage of this



experiment is that it retains high-frequency variabilities in the atmospheric forcing, which are removed in other partially coupled experiments. In addition, this experiment retains the effect of atmospheric feedback after a modification in the

AMOC, which affects the stability of the AMOC (Sherriff-Tadano and Abe-Ouchi 2020). Note that the final results are independent of the choice of the applied atmospheric forcing.

**3 Overall characteristics of MIS3 and MIS5a climates**

The simulated global cooling for MIS3 and MIS5a compared with the Pre-industrial climate (PI) are 5.0 °C and 2.2 °C, respectively (Fig. 2). The strengths of these global surface cooling are smaller compared with that obtained from the LGM

simulation (5.2 °C) with the same model. The simulated MIS3 cooling falls within the range of simulations obtained from previous modelling studies: Guo et al. (2019) simulate the MIS3 climate with the boundary conditions of 38 ka and show a global cooling of 2.9 °C, Merkel et al. (2010) show a cooling of 3.4 °C under 35-ka boundary conditions, Zhang et al. (2014b) show a cooling of 3.5 °C under 38-ka boundary conditions, and Brandefelt et al. (2011) show a global cooling of 5.5 °C under 44-ka boundary conditions. The spatial maps of the surface cooling show a well-known polar amplification pattern

(Fig. 2). In MIS3 and MIS5a, the largest cooling takes place over the North American and Northern Europe as the ice sheets expand southward. In these regions, the surface air temperature drops by more than 10 °C, and is associated with the high albedo and elevation of the ice sheets. The surface cooling is also large over the Southern Ocean, where the local surface cooling exceeds 10 °C and 3 °C for MIS3 and MIS5a, respectively. The surface cooling is relatively mild over the tropics compared with the polar regions, and the areal average cooling over 30°S and 30°N is 3.5 °C and 1.7 °C for MIS3 and

MIS5a, respectively.

The amplified cooling over the polar regions is associated with the expansion of sea ice (Fig. 3). Over the North Atlantic, the Labrador Sea is covered by sea ice in all experiments. Sea ice also expands southward over the Norwegian Sea, though the southern part of the Norwegian Sea still remains ice-free. These expansions in sea ice are consistent with a large surface

cooling simulated in the northern North Atlantic. Over the Southern Ocean, sea ice expands northwards in both experiments compared with the PI. In MIS3, sea ice largely expands northward in the western part of the Southern Ocean and contributes to the large surface cooling observed in that region. In association with the increase in the amount of sea ice, the deep ocean salinity increases and deep ocean temperature decreases in MIS3 compared with PI (Fig. 4). A similar feature is also observed in MIS5a, but with a smaller magnitude. Note that the decrease in ocean temperature is also attributed to the

cooling of the North Atlantic Deep Water (NADW).

Both the MIS3 and MIS5a experiments simulate an interglacial mode of the AMOC (Fig. 5). This is in line with an ice-free condition over the Norwegian Sea and Irminger Sea, where deep water forms due to intense surface cooling (Dokken et al. 2013, Sadazki et al. 2019). Nevertheless, the strength of the AMOC responds differently to MIS3 and MIS5a boundary

forcing (ice sheet, $CO_2$, and insolation). In MIS3, the maximum strength of the AMOC decreases by 3% (−0.5 Sv) and the





AMOC shoals compared with the PI (Fig. 5b). In contrast, the AMOC strengthens in MIS5a by 16% (+2.6 Sv) and shows small changes in depth compared with the PI (16.1 Sv, Fig. 5a). These simulated characteristic of MIS3 and MIS5a are consistent with a reconstruction showing a slightly stronger AMOC in MIS5a and a slightly weaker AMOC in MIS3 (Bohm et al. 2015). Therefore, the simulations of MIS3 and MIS5a capture the large-scale features of climate and deep ocean

circulation reasonably well.

**4 Effect of mid-glacial ice sheet**

**4.1 Global climate and deep ocean circulation**

The results of MIS3-5aice are used to extract the effect of the expansion of mid-glacial ice sheets from MIS5a to MIS3 on the global climate as well as the AMOC. The simulated global cooling in MIS3-5aice is 3.9 °C. This gives a global surface

cooling of 1.1 °C by the expansion of mid-glacial ice sheets (difference between MIS3 and MIS3-5aice) and a global cooling of 1.7 °C by the lowering of $CO_2$ and changes in insolation (difference between MIS3-5aice and MIS5a). The southward expansion of the northern glacial ice sheets induces a large surface cooling over the North America, Northern Europe, and the northern North Atlantic (Fig. 2d). The latter is induced by a vigorous advection of cold air from the North American ice sheet, which expands near the Labrador Sea (Fig. 7b). A slight warming is observed in the Irminger Sea, which is associated

with a shift in the deep-water formation region and sea ice. A surface warming is also observed around Alaska. This is associated with the strengthening of the southerly wind over the eastern North Pacific, which is related to the high surface pressure anomaly over North America induced by the expansion of the glacial ice sheet (Yanase and Abe-Ouchi 2010).

Interestingly, the expansion of the mid-glacial ice sheet exerts an impact on the Southern Ocean by causing a surface cooling

of 2 °C (Fig. 2). This surface cooling is solely induced by the northern mid-glacial ice sheets, because the configuration of the Antarctic ice sheet is fixed to that of the PI. Similar results are reported in Ganopolski and Roche (2009) and Roberts and Valdes (2017). These studies show that the stronger northward oceanic heat transport is responsible for causing the decrease in the surface air temperature over the Southern Hemisphere. Consistent with them, the northward oceanic heat transport is larger in MIS3 compared with MIS3-5aice in our simulations (Fig. 6). This is associated with a cooling of the NADW,

which is induced by the stronger surface cooling by the glacial ice sheets (Fig. 4). As a result, colder deep water outcrops in the Southern Hemisphere and cools the Southern Ocean. Furthermore, associated with the stronger surface cooling over the Southern Ocean, the amount of sea ice also increases in this region (Fig. 3) and increases the deep ocean salinity, which enhances the bottom ocean stratification (Fig. 4).

Unlike the oceanic heat transport, the expansion of mid-glacial ice sheets exerts a very small impact on the AMOC. The maximum strength of the AMOC increases by only 0.5 Sv between MIS3 and MIS3-5aice. These results show that the changes in the AMOC from MIS5a to MIS3 are mostly explained by the modifications to the $CO_2$ levels and insolation. The





small response in the AMOC to ice sheet forcing differs from what is shown by previous studies, which show a strengthening and deepening of the AMOC (Eisenman et al. 2009, Brady et al. 2013, Zhang et al. 2014a, Gong et al. 2015,

Brown and Galbraith 2016, Klockmann et al. 2016, 2018, Kawamura et al. 2017). Analysis of the surface wind stress curl shows an enhancement in response to the mid-glacial ice sheet expansion. This strengthening of surface wind stress curl is mostly explained by the strong northwesterly wind stress anomaly over the Labrador Sea, which is induced by the southward expansion of ice sheets in this region (Fig. 7a, b). As a result of the increased wind stress curl, the wind-driven ocean circulation and the northward transport of salt increases, which tend to intensify the AMOC, as shown by previous studies

(Fig. 7c, d, Montoya and Levermann 2008, Oka et al. 2012, Muglia and Schmittner 2015, Sherriff-Tadano et al. 2018). Nevertheless, the AMOC retains a similar strength in our simulations. This result suggests that other processes are playing a role in compensating for the strengthening effect of the surface wind.

**4.2 Roles of surface cooling by mid-glacial ice sheets on the AMOC**

In addition to surface wind, changes in atmospheric freshwater flux and surface cooling modify the AMOC (Eisenman et al.

2009, Smith and Gregory 2012). The atmospheric freshwater flux can affect the AMOC by modifying the surface salinity field (Eisenman et al. 2009). Figure 8 shows the difference in atmospheric freshwater fluxes between MIS3 and MIS3-5aice. It is found that the expansion of the mid-glacial ice sheet reduces the input of atmospheric freshwater flux over the northern North Atlantic, which is associated with a southward displacement of the westerlies (Hofer et al. 2012), as well as a decrease in specific humidity due to the intense cooling (Laine et al. 2009). This tends to enhance the AMOC by increasing the

surface salinity in the deep-water formation region (Eisenman et al. 2009), which is qualitatively opposite to what we observe now. The strengthening of the surface cooling (Fig. 2d) can cause either a strengthening or weakening of the AMOC by enhancing deep-water formation in the North Atlantic (Oka et al. 2012, Smith and Gregory 2012) or by increasing the amount of sea ice over the northern North Atlantic and Southern Ocean (Oka et al. 2012, Kawamura et al. 2017). Considering the increase in sea ice over both poles (Fig. 3) and the increase in the bottom ocean stratification (Fig. 4),

changes in surface cooling seem to play a role in reducing the AMOC, which is the opposite of the effects of surface wind.

To clarify the effect of surface cooling by the mid-glacial ice sheets on the AMOC, partially coupled (PC) experiments are conducted (Table 2, Fig. 9). In the first two experiments (PC-MIS3 and PC-MIS3-5aice), the surface wind stress and atmospheric freshwater flux are replaced with the climatological forcing of MIS3 and MIS3-5aice, respectively. In both

experiments, while the AMOC becomes stronger than in the corresponding original experiments, it remains in a similar state as simulated in MIS3 and MIS3-5aice. Therefore, the PC experiments reproduce the general pattern of the original experiments. The slight strengthening is associated with an initiation of deep-water formation over the Irminger Sea, which is related to the removal of daily variations in the surface wind (see further discussion in section 4, Figs. 9 and 10).





In the third experiment (PC-MIS3heat), in which the monthly climatology of surface wind stress and atmospheric freshwater
flux from MIS3-5aice are replaced with those of MIS3, the AMOC changes drastically. The maximum strength of the
AMOC decreases to 11 Sv (Fig. 9) and the sea ice covers the deep-water formation region (Fig. 10). Similar weakening is
also observed in PC-MIS3heatano (Fig. 9). Because the surface wind and atmospheric freshwater flux are identical to those
of MIS3-5aice, this result shows that the intense surface cooling by the MIS3 ice sheets reduces the AMOC. These
simulations show that the weakening effect of the surface cooling compensate the strengthening effect of the surface wind,
and hence induces a small change in the AMOC between MIS3 and MIS3-5aice.

How does the intense surface cooling reduce the AMOC? Two processes play a role. The first process is associated with the
intense surface cooling over the northern North Atlantic. Due to this surface cooling, the sea ice increases over the northern
North Atlantic, and it melts over the deep-water formation region and tends to weaken the oceanic convection and the
AMOC (Fig. 10b). In addition, colder water occupies the subsurface ocean in MIS3 compared with MIS3-5aice. As a result,
the oceanic column is more stable with respect to temperature (Fig. 4c). When the mid-glacial ice sheet generates strong
winds, the large surface salinity overcomes the thermally stratified oceanic condition and hence maintains the deep-water
formation. However, when only strong surface cooling is applied, the deep-water formation is interrupted and the AMOC
weakens. The second process involves the Southern Ocean cooling. Due to the intense surface cooling in the northern North
Atlantic, the temperature of the NADW decreases, which is transported to the South and outcrops at the sea surface. This
cooling anomaly is further amplified by the atmosphere and sea ice feedback. As a result, the formation of sea ice near the
Antarctic coastal regions increases and enhances the formation of the Antarctic bottom water (AABW). This causes an
increase of bottom ocean stratification and reduces the AMOC (Buizert and Schmittner 2015, Sun et al. 2016, Klockmann et
al. 2016, 2018). Due to these processes, the AMOC weakens in response to the intense surface cooling by the mid-glacial ice
sheet.

## 5 Discussion

Above results demonstrate a substantial impact of the mid-glacial ice sheets on the global climate. They contribute to a
global cooling of 1.1 °C from MIS5a to MIS3, which is about 40% of the total surface cooling from MIS5a to MIS3. As
shown by previous studies, the expansion of northern glacial ice sheets causes an intense cooling over northern North
America and Europe. Interestingly, the expansion of the mid-glacial ice sheet also causes a surface cooling of 2 °C over the
Southern Ocean. It has been thought that the changes in Northern Hemisphere ice sheet have a small impact on the climate
over the Southern Hemisphere. For example, Manabe and Broccoli (1985) show with an AGCM coupled with a slab ocean
model that the glacial ice sheets have a small impact on the climate over the Southern Hemisphere. The present study shows
that the Northern Hemisphere glacial ice sheets can modify the climate over the Southern Hemisphere and deep ocean via
oceanic heat transport, whose effect is not included in their study. These results hence show the importance of the ocean



dynamics and long integrations in assessing the effect of glacial ice sheet on the global climate. Similar results are also reported for other AOGCMs, which use ice sheet reconstructions of the LGM (Galbraith and de Langen 2019) and deglaciation (Roberts and Valdes 2017). This study further confirms that this effect is applicable in the mid-glacial period as
well.

The changes in ice sheet from MIS5a to MIS3 exert a small impact on the AMOC, unlike the results of previous studies using LGM ice sheets. Partially coupled experiments show that the intense surface cooling by the glacial ice sheets compensates for the strengthening effect of the surface wind by increasing the amount of sea ice over the North Atlantic and
the Southern Ocean. As a result, the induced changes in the AMOC are small. Hence, it is found that the total impact of the expansion of the glacial ice sheet is determined by the balance between the wind effect and the surface cooling effect (Fig. 11). Considering the fact that most climate models show a strengthening of the AMOC in response to the glacial ice sheet expansion, the effect of surface wind dominates in most models.. The reason behind this still remains elusive, though we speculate that it is associated with a strong northerly wind east of the North American ice sheet. Due to this strong northerly
wind anomaly, a large amount of sea ice is transported to the south in MIS3 compared with MIS3-5aice. Thus, the sea ice is transported inefficiently to the deep-water formation region in MIS3. As a result, the cooling effect of the glacial ice sheet may be reduced and thus the wind effect becomes stronger. In contrast, Kim (2004) show a weakening of the AMOC in response to the expansion of the glacial ice sheet. In these simulations, the effect of surface cooling may be stronger than the wind effect. Hence, further analysis on these model outputs would contribute to a better understanding on the relative
importance of wind effect and cooling effect on the AMOC.

Why is the strength of the cooling effect comparable to the wind effect in this study? In the present study, the main difference in the ice sheet between MIS3 and MIS5a appears in the extent of the ice sheet, while the differences in the height is relatively small. According to previous studies, it has been shown that the strength of the wind is largely sensitive to the
height of the ice sheet (Gong et al. 2015, Sherriff-Tadano et al. 2018), while the surface cooling is sensitive to the extent of the ice sheet (Abe-Ouchi et al. 2007). Hence, the changes in surface wind may be smaller compared with other studies using the LGM ice sheet, whereas the change in surface cooling is large in this study. As a result, the AMOC changes only modestly. These results show that the relative strength of the surface wind and surface cooling can depend on the ice sheet configurations, which may cause different responses in the AMOC (Ullman et al. 2014). Furthermore, this result implies that
the history of the shape of the ice sheets (changes in the extent and height) is an important factor when interpreting climate change during the glacial period.

The effect of surface cooling on the AMOC seen here is qualitatively different from some previous studies. For example, Schmittner et al. (2002) show that the strengthening of surface cooling enhances the AMOC in their glacial simulation by
conducting partially coupled experiments with an earth system model of intermediate complexity. In addition, Smith and





Gregory (2012) suggest that the glacial ice sheets enhance the AMOC through strengthening the atmosphere-ocean heat exchange over the deep-water formation region based on their AOGCM experiments. The discrepancy among models may be attributed to two aspects. The first aspect is associated with the strength of surface cooling over the Southern Ocean. If the surface cooling over the Southern Ocean is weak, the weakening effect on the AMOC by glacial ice sheet cooling may be

reduced. The second aspect is related to the thermal threshold of the AMOC. As shown in Oka et al. (2012, 2020), the effect of enhanced surface cooling on the AMOC can depend on the distance from the thermal threshold; when the system is far from the threshold in the parameter space, the surface cooling strengthens the AMOC by enhancing deep-water formation. In contrast, when the system is close to the thermal threshold, stronger surface cooling can cause a drastic weakening of the AMOC. Based on this result, in the present study, the AMOC may be close to the thermal threshold; hence, stronger surface

cooling triggers a drastic weakening of the AMOC, whereas the AMOC may be far from the thermal threshold in other studies. Hence, we do not deny the possibility that the stronger surface cooling can intensify the AMOC when the system is far from the threshold. The important point shown in this study is that the strengthening of surface cooling by glacial ice sheets can affect the thermal threshold of the AMOC and weaken it.

The PC experiments show a slight strengthening of the AMOC compared with the corresponding original experiments. This overestimation is induced by an initiation of deep-water formation over the Irminger Sea, in association with the change in sea ice transport. In the PC experiments, the sea ice becomes thicker near the south-eastern Greenland shore and thinner in the centre of the subpolar region, which increases the gradient of sea ice thickness from the shore to the open ocean (Fig. 10c). Because the climatological surface wind stress applied to the oceanic component is identical between the PC

experiments and original experiments, the only difference in the surface wind stress is the removal of the sub-monthly variations in the surface wind stress in the PC experiments. In fact, in another sensitivity experiment, in which we cyclically apply the raw daily winds of the last 100 years of MIS3 to the oceanic component (PC-MIS3day), the strength of the AMOC and the sea ice thickness resembles that of MIS3 (Figs. 9 and 10d). Hence, the slight increase in the PC experiments are associated with the removal of sub-monthly variations in the surface wind stress, which transport the sea ice from the shore

to the open ocean and reduce the gradient of sea ice thickness. This result implies that the variability in the surface wind on a sub-monthly time-scale plays a role in homogenizing the ice thickness distribution.

The time series of the maximum AMOC shows an increasing trend in PC-MIS3heat (Fig. 9). This trend is associated with two factors: oceanic feedback and the lack of atmospheric feedback in response to a drastic weakening of the AMOC. With

respect to the oceanic feedback, the weakening of the AMOC causes a warming of the subsurface ocean. Furthermore, the warming over the Southern Ocean due to a bipolar see-saw affects the deep ocean stratification (Fig. 11). These processes tend to re-strengthen the AMOC (Brown and Galbraith 2016, Vettoreti and Peltier 2016). With respect to the lack of atmospheric feedback, previous studies show that the expansion of sea ice causes a weakening of the surface wind over the North Atlantic (Byrkedal et al. 2006), which plays a role in maintaining a weak AMOC and extensive sea ice (Figs. 10 and



11, Sherriff-Tadano and Abe-Ouchi 2020). In the partially coupled experiments described above, these atmospheric feedbacks are removed and hence contribute to the destabilization of the weak AMOC. In fact, in PC-MIS3heatano, in which anomalies of the atmospheric forcing are applied, and hence the atmospheric feedback in response to the AMOC weakening is retained, the increasing trend of the AMOC after the weakening is very small (Fig. 9). Therefore, these two processes cause the increasing trend in PC-MIS3heat.


We should note that the volume of the ice sheets used in this study (40-meter sea level equivalent for 80 ka and 96-meter sea level equivalent for 36 ka) are overestimated compared with reconstructions. For example, sea level reconstruction suggests an ice sheet volume of approximately 40- to 90-meter sea level equivalent during MIS3 (Grant et al. 2012, Spratt and Lisiecki 2016, Pico et al. 2017), which is smaller than that used in this study. Furthermore, recent studies even show a much

smaller ice sheet during a portion of MIS3 (Pico et al. 2017, Batchelor et al. 2019). Nevertheless, these reconstructions still show that the ice sheets are slightly larger in MIS3 compared with those in MIS5a (Pico et al. 2017). Hence, while the quantitative effect of the mid-glacial ice sheet might be overestimated in the present study, the qualitative impact of the expansion of MIS3 ice sheet relative to MIS5a is unlikely to change.

The present study has implications for the understanding of climate variability during the glacial period. Ice core studies have shown that the stability and duration of the interstadial climate are strongly related to surface cooling over the North Atlantic (Shultz 2002, Lohmann and Ditlevsen 2019) and Southern Ocean (Buizert and Schmittner 2015).  For example, Shultz (2002) shows that the enhanced surface cooling by the expansion of the glacial ice sheet may explain the shortening of the duration of the interstadial during the mid-glacial period. However, modelling studies have been showing a

strengthening of the AMOC in response to the ice sheet expansion, which stabilizes a vigorous AMOC and the interstadial climate. In contrast, this study shows a possibility that the expansion of glacial ice sheet can in fact cause the weakening of the AMOC and hence destabilization of the vigorous AMOC, which contributes to shorter interstadial duration. This is the case when the effect of surface cooling dominates the effect of surface wind. In this case, both the enhanced surface cooling over the North Atlantic and the Southern Ocean can play a role, which are consistent with ice core studies (Buizert and

Schmittner 2015, Lohmann and Ditlevsen 2019). This result suggests that the changes in the relative strength of surface wind and surface cooling can affect the AMOC and climate drastically and may be important in interpreting the millennial time-scale climate variability of the glacial periods.

**6 Conclusions**

In this study, the role of mid-glacial ice sheets on the global climate and the AMOC is explored. For this purpose,

simulations of MIS3 and MIS5a are conducted with the comprehensive climate model MIROC4m. The ice sheet configurations are taken from an ice sheet model, which reproduces the ice sheet evolution over the past 400,000 years (Abe-



Ouchi et al. 2013). Furthermore, to assess the processes by which the mid-glacial ice sheets affect the AMOC, PC experiments are conducted with MIROC4m. The main results of the present study can be summarised as follows:

- In the MIS3 and MIS5a simulations, the global average temperature decreases by 5.0 °C and 2.2 °C, respectively, compared with the PI climate simulation. Comparison of the MIS3 and MIS5a results show that the expansion of mid-glacial ice sheets contributes to a global cooling of 1.1 °C, which is about 40% of the total surface cooling from MIS5a to MIS3 of about 2.8 °C.

- The southward expansion of northern mid-glacial ice sheets not only causes a drastic cooling over the northern North Atlantic, but also causes a 2 °C of surface cooling over the Southern Ocean. The cooling over the Southern Ocean is associated with the cooling of the NADW.

- The AMOC is enhanced by 16% in MIS5a, whereas it weakens by 3% in MIS3 compared with the preindustrial climate simulation. The weaker AMOC in MIS3 compared with MIS5a is consistent with a previous proxy study (Bohm et al. 2015).

- A sensitivity experiment that modifies the glacial ice sheets showed that the southward ice sheet expansion exerts a very small impact on the AMOC (0.5 Sv), despite the strengthening of surface wind and wind-driven ocean circulation over the North Atlantic, which tends to intensify the AMOC (Oka et al. 2012, Klockmann et al. 2016, Sherriff-Tadano et al. 2018).

- Partially coupled experiments reveal that the intense surface cooling by the glacial ice sheet weakens the AMOC and counteracts the strengthening effect of surface wind. The surface cooling increases the sea ice over the northern North Atlantic, which melts over the deep-water formation region and weakens oceanic convection. Also, the northern surface cooling causes a cooling over the Southern Ocean, which strengthens the AABW, increases the stratification of the bottom ocean, and weakens the AMOC.

- It is found that the total impact of glacial ice sheets on the AMOC is determined by the relative strength of two factors, surface wind and surface cooling. In most models, the effect of surface wind is stronger; hence, the AMOC strengthens in response to the ice sheet expansion. In the present study, the main difference in the ice sheets appears in their extent, rather than their height. As a result, the strength of the cooling effect become comparable to that of the wind effect, and it causes small changes in the AMOC. Our result suggests that the relative strength of the wind effect and cooling effect depends on the shape of the ice sheet reconstructions.

The results of the present study also offer a global dataset of climate during MIS3 and MIS5a, which is still lacking compared with the LGM (Gong et al. 2013, Guo et al. 2019). Recently, the amount of reconstruction of mid-glacial period is increasing (Jensen et al. 2018), and hence more detailed comparisons with these reconstructions need to be conducted in future. Furthermore, the present results provide a reference climate state for investigating the millennial time-scale climate variability that occurred during the mid and early glacial period (Henry et al. 2016, Mitsui and Crucifix 2017, Guo et al. 2019). In a forthcoming study, we will perform freshwater hosing experiments with these simulations and investigate how the changes in boundary conditions affect climate variability and the recovery time of



the AMOC. This can contribute to a better understanding of millennial time-scale climate variability, which is still not fully understood and remains as one of the largest questions in the study of paleoclimate.

**Code and data availability**

The code of MIROC associated with this study is available to those who conduct collaborative research with the model users
under license from copyright holders. The code of partial couple experiments is available from corresponding author (S. S.-T.) upon reasonable request. The simulation data will be available from https://ccsr.aori.u-tokyo.ac.jp/~tadano/.

**Author contribution**

All authors conceived the study. S. S.-T. performed the climate model simulation and analyzed the results with the assistance of A. A.-O., and A. O.. S. S.-T. performed the partially coupled experiments with the assistance of A. O.. Manuscript written
by S. S.-T. with contributions from all authors.

**Competing interest**

The authors declare no competing interests.

**Acknowledgements**

We thank Masahide Kimoto, Hiroyasu Hasumi, Masahiro Watanabe, Ryuji Tada, and Takashi Obase for constructive
discussion. The model simulations were performed on Earth Simulator 3 at JAMSTEC. This study was supported by Program for Leading Graduate Schools, MEXT, Japan, and JSPS KAKENHI Grant Number 15J12515, 17H06104, 17H06323.

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





**Table 1: Forcing and boundary conditions of climate simulations. Results of global mean temperature (GMT) and Atlantic meridional overturning circulation (AMOC) are also shown.**

| Name | $CO_2$ | Ice sheet | Obliquity | Precession | Ecc | GMT | AMOC |
|------|--------|-----------|-----------|------------|-----|-----|------|
| MIS5a | 240 ppm | 80 ka | 23.175 | 312.25 | 0.0288 | 10.58˚C | 18.7 Sv |
| MIS3 | 200 ppm | 36 ka | 22.754 | 251.28 | 0.0154 | 7.85˚C | 15.6 Sv |
| MIS3-5aice | 200 ppm | 80 ka | 22.754 | 251.28 | 0.0154 | 8.91˚C | 15.1 Sv |

**Table 2: Partially coupled experiments. In PC-MIS3heatano, climate anomalies in surface wind and atmospheric freshwater (FW) flux between MIS3-5aice and MIS3 are added to MIS3.**

| Name | Surface wind | Atmos. Fw flux | Surface cooling |
|------|--------------|----------------|-----------------|
| PC-MIS3 | MIS3 | MIS3 | MIS3 |
| PC-MIS3-5aice | MIS3-5aice | MIS3-5aice | MIS3-5aice |
| PC-MIS3heat | MIS3-5aice | MIS3-5aice | MIS3 |
| PC-MIS3heatano | MIS3-5aice | MIS3-5aice | MIS3 |


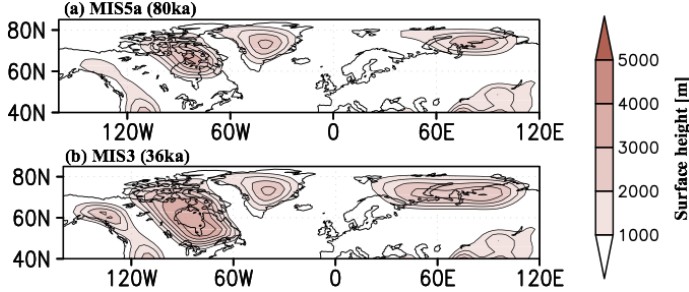

**Figure 1: Topography of (a) MIS5a (80 ka) and (b) MIS3 (36 ka). Results from an ice sheet model are presented (Abe-Ouchi et al. 2013). These ice sheet configurations are used for climate model simulations.**





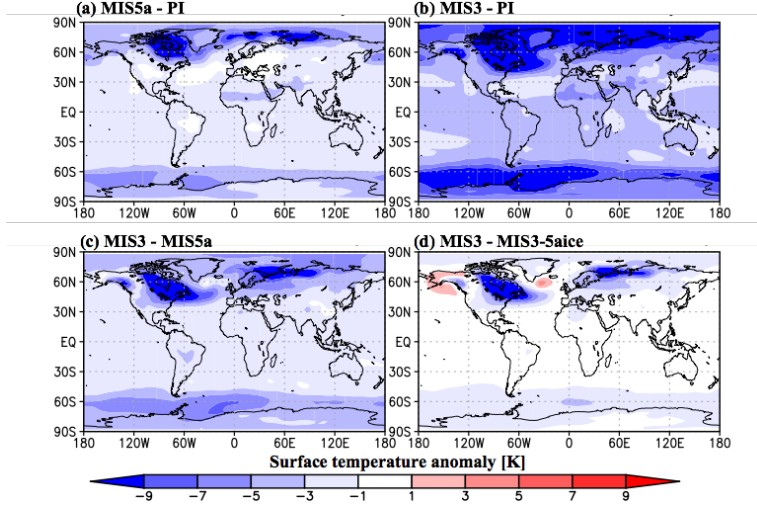


**Figure 2: Surface air temperature anomalies calculated from the AOGCM. The 100-year climatology is used to calculate the anomalies. (a) MIS5a minus PI and (b) MIS3 minus PI. In (c), differences between MIS3 and MIS5a are shown. In (d), the effect of ice sheet expansion from MIS5a to MIS3 is shown (MIS3 minus MIS3-5aice).**

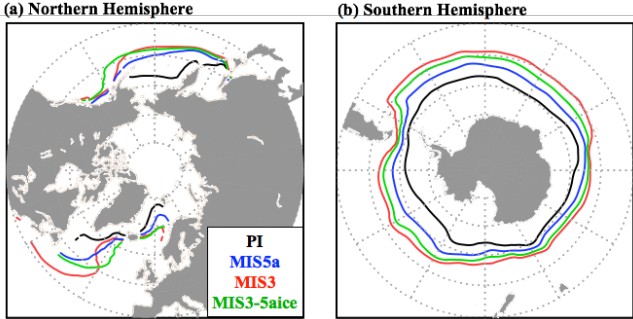

**Figure 3: Annual mean sea ice coverage simulated from the AOGCM. The coverage is defined by 15% sea ice concentration. A 100-year average is used. (a) Northern Hemisphere, (b) Southern Hemisphere. Black: PI, Blue: MIS5a, Red: MIS3, and Green: MIS3-5aice.**





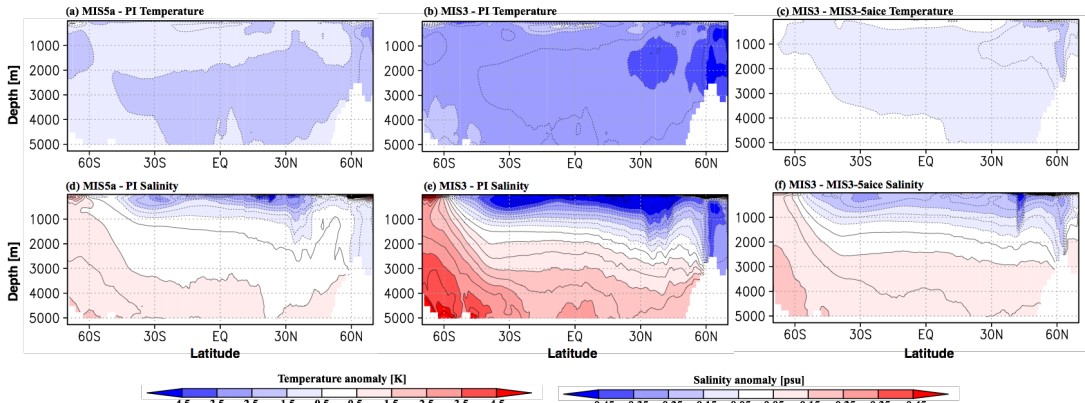

**Figure 4: Anomalies of zonally averaged oceanic properties over the Atlantic simulated from the AOGCM. The top panels show**
**temperature anomalies and the bottom panels show salinity anomalies. (a, d) MIS5a minus PI, (b, e) MIS3 minus PI, (c, f) MIS3 minus MIS3-5aice. The climatology of the last 100 years is used to create these figures.**

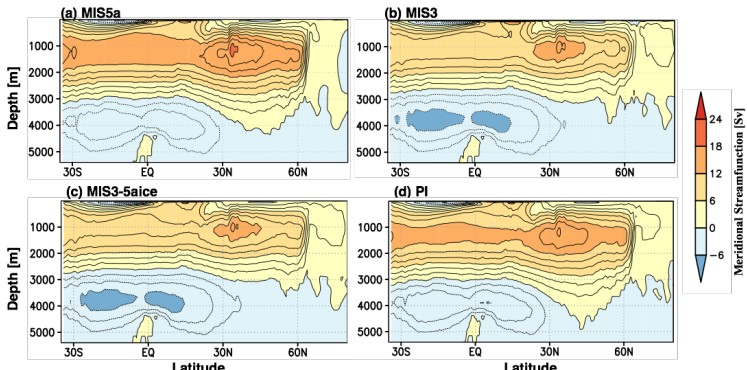

**Figure 5: Meridional streamfunction (Sv=10$^6$ m$^3$ s$^{-1}$) over the Atlantic simulated from the AOGCM. (a) MIS5a, (b) MIS3, (c)**
**MIS3-5aice, and (d) PI. The climatology of the last 100 years is used to create these figures.**





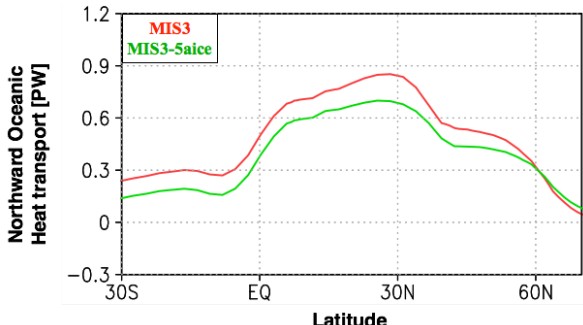

**Figure 6: Northward oceanic heat transport over the Atlantic basin simulated from the AOGCM. Red: MIS3 and Green: MIS3-5aice. The climatology of the last 100 years is used to create these figures.**

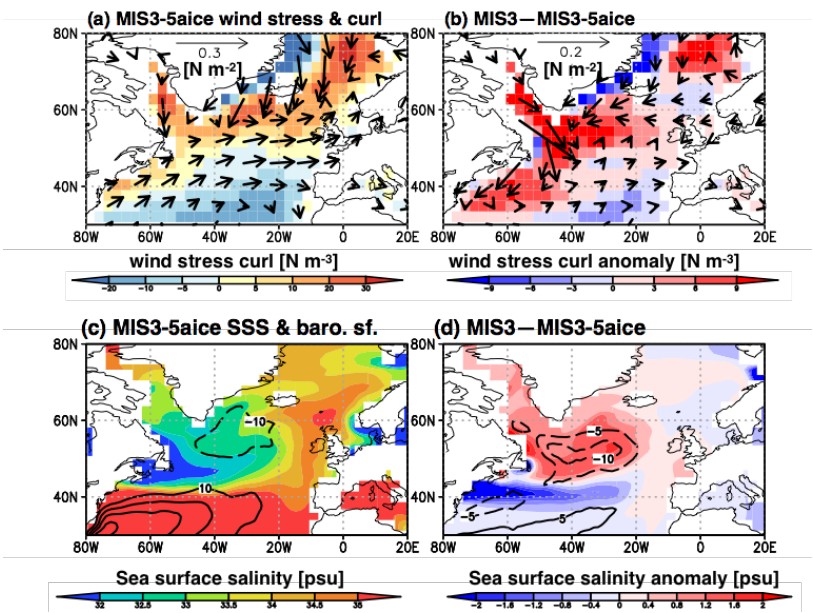

**Figure 7: Changes in surface wind stress and wind-driven ocean circulation associated with expansion of the northern glacial ice sheet from MIS5a to MIS3. Top figures show the surface wind stress (arrow, N m$^{-2}$) and wind stress curl (colour, N m$^{-3}$), and bottom figures show the barotropic streamfunction (contour, Sv) and sea surface salinity (colour, psu). (a, c) MIS3-5aice and (b, d) MIS3 minus MIS3-5aice (ice sheet effect). The climatology of the last 100 years is used to create these figures.**



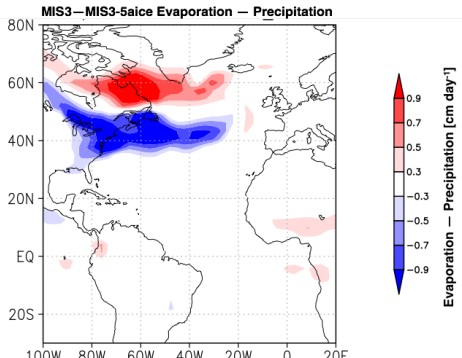

**Figure 8:** Anomalies of atmospheric freshwater flux (E-P, cm day⁻¹) out of the ocean between MIS3 and MIS3-5aice. Red colour shows freshwater flux out of the ocean and blue colour shows freshwater flux into the ocean. The climatology of the last 100 years is used to create these figures.

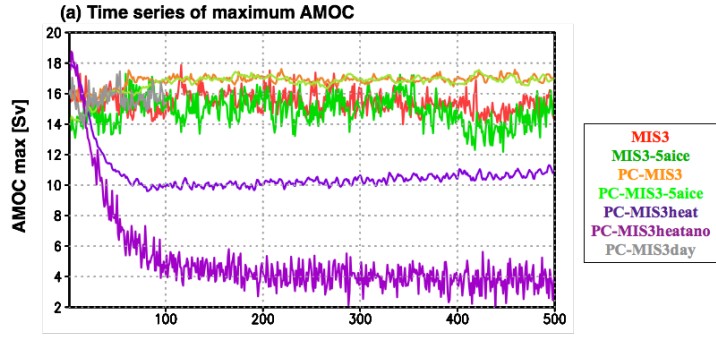

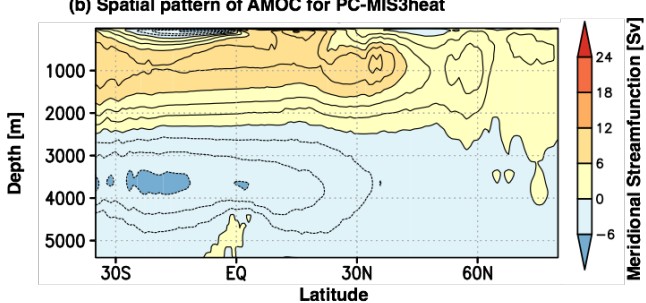





**Figure 9: Results of partially coupled experiment conducted with the AOGCM. (a) Time series of the maximum strength of the**
**AMOC. (b) Spatial pattern of the Atlantic meridional streamfunction calculated from PC-MIS3heat. The climatology of the last**
**100 years is used to create this figure.**

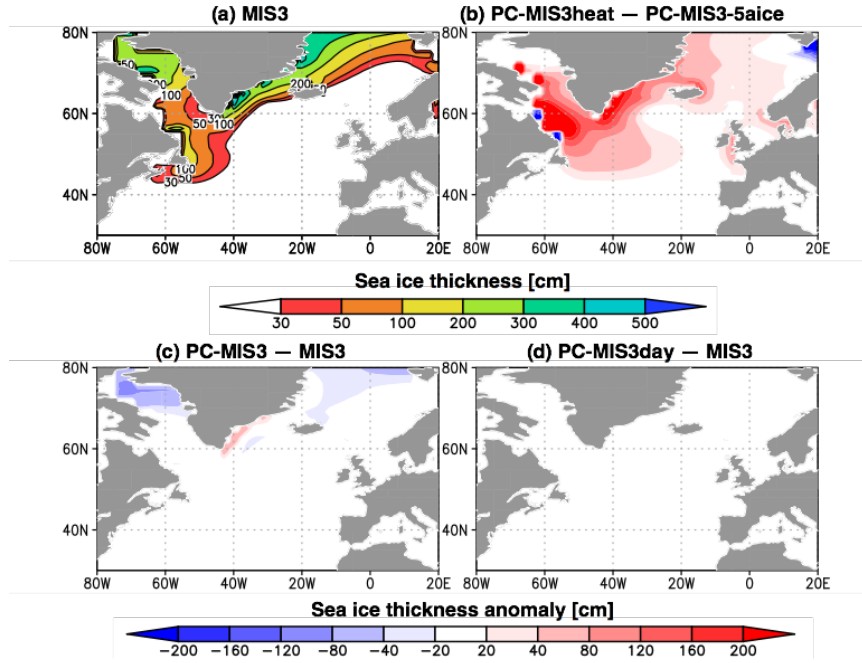

**Figure 10: Annual mean sea ice thickness (cm, colour) over the North Atlantic simulated from the AOGCM and partially coupled**
**experiments. (a) MIS3. (b) Effect of surface cooling by mid-glacial ice sheet (PC-MIS3heat minus PC-MIS3-5aice). (c) and (d)**
**show the reproducibility of sea ice thickness by the partially coupled experiment: (c) PC-MIS3 minus MIS3 and (d) PC-MIS3day**
**minus MIS3. In (a), (b), and (c), the results of the last 100 years are used. In (d), the results of the last 50 years are used for PC-**
**MIS3day.**





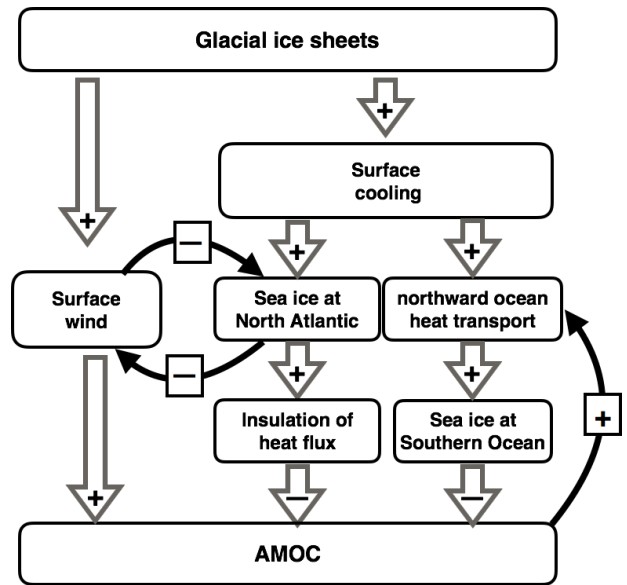

**Figure 11: Schematic of the processes by which changes in the glacial ice sheet affect the AMOC. The black solid arrows indicate a**

**feedback within the atmosphere-sea ice-ocean system (Kawamura et al. 2017, Sherriff-Tadano and Abe-Ouchi 2020).**