# Peer review of "Impact of mid-glacial ice sheets on deep ocean circulation and global climate"

_Climate of the Past, 2020_

## Referee Comment (RC1) · Chuncheng Guo (Referee) · 2 Jul 2020

Sam Sherriff-Tadano and co-authors conducted an AOGCM study investigating the impact of an expanded North American mid-glacial ice sheet on the ocean circulation. By using partially coupled experiments, they found that an ice sheet-induced cooling in the North Atlantic and Southern Ocean can lead to a weakening of the AMOC in their model, which competes with the strengthening effect of an enhanced wind forcing. The overall effect is a relatively small change in the strength of AMOC during MIS3 compared to the pre-industrial. The authors examined in detail the dynamics and processes at play with sensitivity and partially coupled experiments.

I find the study very interesting and I am overall positive on the manuscript. It fits the scope of the journal, and would serve as a useful reference for the community that work on understanding the millennial-scale climate variabilities in the last glacial period. However, I do have some comments which I hope can help improve the manuscript.

Major comments:

>L24-34: I suggest that the authors add a schematic figure illustrating the time evolution of certain climate variables from paleo records (e.g. summer insolation, CO2, sea level, d18O etc.) from last interglacial to the present day. This can provide a more clear context and would be especially beneficial to a wider audience.

>section 2.1: Could the authors add a short paragraph briefly summarizing the performance of MIROC4m for the preindustrial and/or present day simulations, especially for the metrics that are relevant for the analysis later in the main text? Such metrics can include, but not limited to, sea ice concentration, mixed layer depth, ocean profiles/stratification in the North Atlantic. Climate sensitivity would be useful to mention as well. Any significant bias and therefore its implication for the conclusions drawn in this work should also be discussed where relevant.

>L121: Did the authors perform any sensitivity experiment with regard to the opening/closing of the Bering Strait by any chance? If yes would be useful to briefly discuss it here. Some studies have shown how an opened/closed Bering Strait could have some significant impact on the North Atlantic ocean state.

>L355-357: It is not immediately clear to me how do subsurface warming and southern ocean warming are able to re-strengthen the AMOC. The latter due to reduced production of AABW? How about subsurface warming? Please elaborate a bit more on the dynamic links here.

>L358-359: Once again, it is not clear to me the link between the expanded sea ice and a weakening surface wind. My understanding is that a more extensive sea ice cover in

the North Atlantic 'protects' the ocean surface from the wind stress above, which tends to spin down the ocean circulation, and is favorable for maintaining a weak AMOC.

>L340-364: I appreciate the authors' efforts in explaining some of the interesting modelling results here. However, to me this part has a very limited contribution to the main points of the paper, and could be a distraction to the readers in this section. I think by removing it or moving it to supplementary material could help enhance the legibility of this section. It is up to the authors to decide though.

>Fig. 11: this schematic is not adequately discussed/referred to in the main text. There are several places in the text (mainly in 'Discussion') where the relevant processes are described and should refer to this figure. In addition, the feedbacks indicated by the black solid arrows are not straightforward to me. Please consider elucidating it more explicitly in the main text or in the caption where appropriate.

Minor and technical comments:

>title: I think that it is good practice to try to avoid abbreviations in the title (e.g. AMOC).

>L8: should spell out that it is about the expansion of ice sheet in North America.

>L10: it would be useful to mention the MIS3 and 5a time slices that the authors chose in this study, such that the readers can get a quick grasp by reading the abstract.

>L55-59: suggest to rephrase the sentence as "…, which can cause either a strengthening of the AMOC by …, or a weakening of the AMOC by…" This also applies to L246-248.

>L65: you mean "For" these two periods?

>L71: "…, whose effect of surface cooling is prominent." This reads a bit ambiguous to me; please consider rephrasing it.

>L108-109: is it relevant to include the information in the square bracket? If not please consider removing it.

>L146-147: To my understanding, it should be stressed that surface heat flux cannot be imposed because it is strongly coupled to SST, whereas surface freshwater flux can because there is no direct SSS feedback to the flux.

>L168-170: I am a bit surprised the simulated LGM climate is only about 0.2 deg-C colder than the MIS3 climate, considering that there is a $CO_2$ difference of 20 ppm plus some (supposedly moderate) difference in the distribution of ice sheet. Could the authors comment on this?

>L184: perhaps the reference of Dokken et al. and Sadazki et al. in lines 193-194 can be moved here.

>L186: I find it a bit odd to say "the western part of the Southern Ocean"; suggest to change to, for example, Pacific/Indian/Atlantic Ocean sector of the Southern Ocean.

>L220: it is not clear from Fig. 4 that there is 'stronger surface cooling'. I see a relatively homogenous distribution of ocean cooling in Fig 4(a,b). Is this the case or it has to do with the color bar?

>L222: "and increases the deep ocean salinity, . . ." error in grammar. Also, should spell out the increased deep ocean salinity is via brine rejection.

>L235: suggest to move "Fig. 7c,d" to the middle of L234.

>L261: change "are replaced with" to "replace with"?

>L265: "compensates"

>L269-271: "Due to . . . AMOC (Fig. 10b)." To me the main effect of sea ice in weakening the AMOC in the north Atlantic is because of its insulation that reduces air-sea flux and therefore ocean convection. The effect of melting of sea ice, if one can do a back-of-envelope calculation converting the melted sea ice into sverdrups, should be relatively small.

>L272: again, the more stable ocean column is not clear to me from Fig. 4c.

>L273: suggest to tone down "overcomes" to "tends to overcome".

>L283: "The results above demonstrate..."?

>L303: there are two full stops.

>L303-307: this reads very speculative to me, if I understand the authors' point correctly here. Please consider removing it or providing more evidence (it's up to the authors to decide).

>L329: ice sheet"-induced" cooling?

>L335: replace "deny" with "exclude"?

>L348: "resemble"?

>Fig. 9: the color of "PC-MIS3-5aice" in the legend is not correct.

―――――――――――――――――――――

---

## Referee Comment (RC2) · Anonymous Referee #2 · 10 Jul 2020

Sherriff-Tadano et al. has presented a study about the impact of mid-glacial ice sheet expansion on Atlantic Meridional Overturning Circulation via changing surface winds and surface cooling, based on fully and partially coupled experiments using MIROC model. They found that the relative strength of surface wind and surface cooling depends on the ice sheet configuration, and the strength of the surface cooling can be comparable to that of surface wind when changes in the extent of ice sheet are prominent. In the manuscript, the authors have discussed their results based on a nice review about the existing studies, meanwhile some parts of their own results need to be provided to support their conclusions. In the below, here is listed my comments:

[Figure]

Major comments: 1. In Figure 1, please add a panel for the ice-sheet anomalies between 36ka and 80ka, since it is a key to interpolate the modelling results.

2. Line 116: why to use the CO2 concentration and insolation at 35ka, instead of 36ka? A linguistic error? Or specific reason?

3. Line 170: Please add a reference for the LGM experiment.

4. Line 195-196: Please give the value of AMOC strength in PI experiment.

5. Line 219 and Figure 6: Please add the curve for the modelled PI state.

6. In what area are the NADW formed? Are they consistent among experiments? Any response of the NADW formation in the NORDIC Sea?

7. Line 249: bottom ocean stratification with respective to density? If so, please add the information for density in Figure 4.

8. Line 271: In addition to Figure 10, please show the convection map as that in Fig. 7c.

9. Also in Line 271: please add a figure for the statement 'colder water occupies the subsurface ocean in MIS3 compared with MIS3-5aice.' , in either Main text or SI.

10. In Table 1: please add the information also for the PI and LGM experiments together with their references.

11. In Figure 11, how to address the impact of stronger surface winds on the northward ocean heat transport and surface cooling in the northern North Atlantic? Any indications based on the experiments in this study?

Minor comments: Line 37: 'Project' to 'Projects'

Line 210: please refer to Figure 2d, for the warmer surface around Alaska

Line 303: '.' has been double used.

---

## Author Comment (AC1) · 2 Sep 2020

Reply to Reviewer1

We are grateful to the reviewer for his time in evaluating the manuscript and his constructive comments and suggestions. As listed below, we have taken all the comments into account by the reviewer in the revised manuscript. In the following, our responses will be written in blue, while the comments by the reviewer will be written in black.

>L24-34: I suggest that the authors add a schematic figure illustrating the time evolution of certain climate variables from paleo records (e.g. summer insolation, CO2, sea level, d18O etc.) from last interglacial to the present day. This can provide a more clear context and would be especially beneficial to a wider audience.
We agree to the suggestion. We will add a figure illustrating the time evolution of summer insolation, CO2, sea level, Greenland ice core data, and AMOC in Fig.1 of the revised manuscript.

[Figure]

Figure 1: Time series of climate records of the last glacial period. (a) 65˚N July insolation (W m$^{-2}$), (b) black: sea level data from Spratt and Lisiecki (2016), brown: sea level data from Grant et al. (2012), gray: simulated time evolution of ice sheet (Abe-Ouchi et al. 2013), (c) CO$_2$ (Bereiter et al. 2015), (d) Greenland ice core delta 18 O from North Greenland Ice Core Project (NGRIP) core (Rasumussen et al. 2013). (e) Bermuda Rise $^{231}$Pa/$^{230}$Th (Bohm et al. 2015), which is a proxy of the strength of the AMOC. Red and Blue shades correspond to the simulated period of MIS3 and MIS5a in our climate model simulations, respectively.

>section 2.1: Could the authors add a short paragraph briefly summarizing the performance of MIROC4m for the preindustrial and/or present day simulations, especially for the metrics that are relevant for the analysis later in the main text? Such metrics can include, but not limited to, sea ice concentration, mixed layer depth, ocean profiles/stratification in the North Atlantic. Climate sensitivity would be useful to mention as well. Any significant bias and therefore its implication for the conclusions drawn in this work should also be discussed where relevant.

Following the reviewer's suggestion, we will add a following paragraph in the method section in the revised manuscript.

"The model version used in this study reproduces the modern AMOC (Fig. 6d), the deepwater formation over the Nordic Seas (Fig. S1) and sea ice extent over the North Atlantic (Fig. 4) reasonably well as in the previous version (Otto-Bliesner et al. 2007, Weber et al. 2007, Kawamura et al. 2017). While the current model version overestimates sea ice extent and lacks deepwater formation over the Labrador Sea (Fig. S1, Fig. 4), the performance of the modern Southern Ocean sea ice extent has improved compared with the previous version (Fig. 4). This model version has also been used extensively for paleoclimate (Obase and Abe-Ouchi 2019) and future climate studies (Yamamoto et al. 2015). It has a climate sensitivity of 4.1 K and reproduces the AMOC of the LGM reasonably well (Sherriff-Tadano and Abe-Ouchi 2020)"

>L121: Did the authors perform any sensitivity experiment with regard to the opening/closing of the Bering Strait by any chance? If yes would be useful to briefly discuss it here. Some studies have shown how an opened/closed Bering Strait could have some significant impact on the North Atlantic ocean state.

As pointed out by the reviewer, the closure of the Bering Strait can have an impact on the AMOC. However, unfortunately, we have not performed sensitivity experiments closing the Bering Strait. Nevertheless, we will add a following sentence to the revised manuscript so that the readers can refer to the effect of Bering Strait on the AMOC in other studies.

"The global sea level is unchanged, and the land sea mask outside the northern glacial ice sheet region is same as the modern configuration (e.g., the Bering Strait remains open, which itself may impact on the AMOC (Hu et al. 2015))."

>L355-357: It is not immediately clear to me how do subsurface warming and southern ocean warming are able to re-strengthen the AMOC. The latter due to reduced production of AABW? How about subsurface warming? Please elaborate a bit more on the dynamic links here.

We removed the sentence associated with the subsurface warming since we agree that the accumulation of heat at the subsurface ocean over the North Atlantic does not cause a gradual recovery of the AMOC, but rather causes an abrupt strengthening by triggering a new deepwater formation. With respect to the Southern Ocean process, we will increase the explanation and also add some references, which show the effect of temperature changes over the Southern Ocean on the AMOC (Buizert and Schmittner 2015, Jansen 2017), to support our discussion. We also nuanced the paragraph since the original sentence looked too confident.

"With respect to the oceanic feedback, the weakening of the AMOC causes a warming over the Southern Ocean due to the reduction in the northward heat transport, and hence reduces the deep ocean stratification and the AABW (Fig. 11). These processes can contribute to re-strengthen the AMOC (Buizert and Schmittner 2015, Jansen 2017). "

>L358-359: Once again, it is not clear to me the link between the expanded sea ice and a weakening surface wind. My understanding is that a more extensive sea ice cover in the North Atlantic 'protects' the ocean surface from the wind stress above, which tends to spin down the ocean circulation, and is favorable for maintaining a weak AMOC.

As the reviewer says, the link between the expanded sea ice and a weakening of the surface wind was not clear in the original manuscript. In the revised manuscript, we increased the description on this topic as follows;

"With respect to the lack of atmospheric feedback, previous studies show that the expansion of sea ice causes a weakening of the surface wind over the North Atlantic by increasing the static stability of the lower troposphere (Byrkedal et al. 2006, Sherriff-Tadano and Abe-Ouchi 2020). They further show that this weakening of the surface wind plays a role in maintaining a weak AMOC by reducing the wind-driven transport of salt to the deepwater formation region (Zhang et al. 2014a, Sherriff-Tadano and Abe-Ouchi 2020)."

Also, as the reviewer pointed out, the extensive sea ice protects the ocean surface from the wind stress, and causes a weakening of the wind-driven ocean circulation. In fact, this feedback is taken into account in the model experiments (both the original and partially coupled) when the sea ice expands. However, if this feedback is dominant, one should expect a stable weak AMOC, rather than a gradual increase in the AMOC, which is observed in our partially coupled experiments (Fig. 10). This result suggests that other processes/feedback after the weakening of the AMOC is causing the gradual increase in the AMOC. From several previous studies presented above, we speculate that the lack of the sea ice-wind feedback can destabilize the weak AMOC and cause the gradual increase in the AMOC.

>L340-364: I appreciate the authors' efforts in explaining some of the interesting modelling results here. However, to me this part has a very limited contribution to the main points of the paper, and could be a distraction to the readers in this section. I think by removing it or moving it to supplementary material could help enhance the legibility of this section. It is up to the authors to decide though.

Following the reviewer's suggestion, we decided to move the first paragraph to the supplement to increase the eligibility of the section. Related to this, we moved Fig. 10c,d in the original manuscript to Fig. S3 in the revised manuscript because it is no longer discussed in the main manuscript. For the second paragraph, we decided to keep it in the main manuscript since we think this paragraph discusses important internal feedback within the atmosphere-ocean system, which can appear in the simple schematic figure.

>Fig. 11: this schematic is not adequately discussed/referred to in the main text. There are several places in the text (mainly in 'Discussion') where the relevant processes are described and should refer to this figure. In addition, the feedbacks indicated by the black solid arrows are not straightforward to me. Please consider elucidating it more explicitly in the main text or in the caption where appropriate.

In the revised manuscript, we will adequately refer to the schematic figure (Fig. 12 in the revised manuscript) in 4.2, 5, and 6. In addition, we increased the explanation of the feedbacks as described above. Also, in order to concentrate on our main finding, we decided to remove the black arrows of internal feedback from the revised Fig. 12. Nevertheless, we modified the caption so that the reader can refer to the main text for the discussion on the possible internal feedbacks.

[Figure]

Figure 12: Simple schematic of the processes by which changes in the glacial ice sheet affect the AMOC. Possible internal feedbacks within the atmosphere-sea ice-ocean system are discussed in the Discussion section.

Minor and technical comments:
>title: I think that it is good practice to try to avoid abbreviations in the title (e.g. AMOC).
Corrected.

>L8: should spell out that it is about the expansion of ice sheet in North America.
Corrected.

>L10: it would be useful to mention the MIS3 and 5a time slices that the authors chose in this study, such that the readers can get a quick grasp by reading the abstract.
We will add the time slice in the revised manuscript as well as in the revised Fig. 1.

>L55-59: suggest to rephrase the sentence as ": : :, which can cause either a strengthening of the AMOC by : : :, or a weakening of the AMOC by: : :" This also applies to L246-248.
Corrected.

>L65: you mean "For" these two periods?
We meant that the ice sheet is considered to be slightly larger in MIS3 compared with MI5a.

>L71: ": : :, whose effect of surface cooling is prominent." This reads a bit ambiguous to me; please consider rephrasing it.
We will modify this sentence in the revised manuscript as follows;
":::, whose effect of the ice sheet extent and hence the surface cooling is prominent."

>L108-109: is it relevant to include the information in the square bracket? If not please consider removing it.
We think this sentence is important to avoid readers from getting confused when they look back to previous articles. Therefore, we will remain this sentence.

>L146-147: To my understanding, it should be stressed that surface heat flux cannot be imposed because it is strongly coupled to SST, whereas surface freshwater flux can because there is no direct SSS feedback to the flux.

As suggested by the reviewer, we modified the sentence as follows,

"Following previous studies (Schmittner et al. 2002, Gregory et al. 2005), the heat flux is unchanged in these experiments. This is because the heat flux is strongly coupled to the sea surface temperature and that fixing the surface heat condition has an unrealistic impact on the AMOC (Marozke 2012)."

>L168-170: I am a bit surprised the simulated LGM climate is only about 0.2 deg-C colder than the MIS3 climate, considering that there is a CO2 difference of 20 ppm plus some (supposedly moderate) difference in the distribution of ice sheet. Could the authors comment on this?

We were also surprised at the result when we saw it. One possible reason is the lower obliquity in this experiment compared with the LGM. Several previous studies have shown that the lower obliquity can cause an annual global cooling by increasing the sea ice in the Southern Ocean and Arctic regions, even though the global input of insolation does not differ. We added a following sentence on this point in the revised manuscript.

"The strong MIS3 cooling similar to that of LGM is possibly related to the low obliquity applied in MIS3, which increases the amount of sea ice in both hemispheres and causes a global cooling through feedbacks within the atmosphere-ocean coupled system (Galbraith and de Lavergne 2019). "

>L184: perhaps the reference of Dokken et al. and Sadazki et al. in lines 193-194 can be moved here.

We added these reference in the sentence.

>L186: I find it a bit odd to say "the western part of the Southern Ocean"; suggest to change to, for example, Pacific/Indian/Atlantic Ocean sector of the Southern Ocean.

Corrected.

>L220: it is not clear from Fig. 4 that there is 'stronger surface cooling'. I see a relatively homogenous distribution of ocean cooling in Fig 4(a,b). Is this the case or it has to do with the color bar?

As pointed out by the reviewer, the original sentence was misleading. We should have clearly mention that we are referring to Fig. 4c, which shows the effect of ice sheet on ocean temperature. We have modified this sentence as follows,

"This is associated with a cooling of the NADW (Fig. 4c), which is induced by the stronger surface cooling by the glacial ice sheets"

In Fig.4c, you can find a cooling of NADW, which is associated with the ice sheet expansion and hence the resulting stronger surface cooling.

>L222: "and increases the deep ocean salinity, : : :" error in grammar. Also, should spell out the increased deep ocean salinity is via brine rejection.

Corrected. Also added the explanation of brine rejection.

>L235: suggest to move "Fig. 7c,d" to the middle of L234.

Corrected.

>L261: change "are replaced with" to "replace with"?
As pointed out by the reviewer, this sentence was strange. We modified the sentence as follows.
"In the third experiment (PC-MIS3heat), in which the monthly climatology of surface wind stress and atmospheric freshwater flux of MIS3 are replaced with those of MIS3-5aice"

>L265: "compensates"
Corrected.

>L269-271: "Due to : : : AMOC (Fig. 10b)." To me the main effect of sea ice in weakening the AMOC in the north Atlantic is because of its insulation that reduces air-sea flux and therefore ocean convection. The effect of melting of sea ice, if one can do a back-of-envelope calculation converting the melted sea ice into sverdrups, should be relatively small.
As the reviewer suggests, the expansion of sea ice weakens the AMOC by suppressing the atmosphere-ocean heat exchange (Oka et al. 2012). In addition, it has been shown that the increase in sea ice over the north North Atlantic can reduce the AMOC and the ocean convection via meltwater at the sea ice edge (Born et al. 2010). Following these previous studies, we will modify this sentence as follows;
"Due to this surface cooling, the sea ice increases over the northern North Atlantic (Fig. 11b). The increase in sea ice tends to weaken the oceanic convection and the AMOC by insulating the atmosphere-ocean heat flux (Oka et al. 2012) and by increasing the meltwater flux over the deep-water formation region and (Born et al. 2010). "

>L272: again, the more stable ocean column is not clear to me from Fig. 4c.
We will add a figure of vertical profile of ocean temperature in the Fig. S2, which shows that MIS3 exhibits more stable ocean column in terms of temperature compared with MIS3-5aice.

[Figure]

Figure S2: Vertical profile of oceanic properties at the North Atlantic Deep Water formation region (60˚W-0˚, 55˚N-65˚N). Red: MIS3 and Green: MIS3-5aice. Cold water occupies the subsurface ocean in MIS3 compared with MIS3-5aice. The climatology of the last 100 years is used to create these figures.

>L273: suggest to tone down "overcomes" to "tends to overcome".
Corrected.

>L283: "The results above demonstrate: : :"?
Corrected.

>L303: there are two full stops.
Corrected. Thanks for pointing out.

>L303-307: this reads very speculative to me, if I understand the authors' point correctly here.
Please consider removing it or providing more evidence (it's up to the authors to decide).
Indeed it is a speculative discussion, but we think this point is quite important, which the modelers
should keep in mind. We will add a figure supporting this sentence in the supplementary file and
keep this discussion in the revised manuscript.

[Figure]

Figure S4: Spatial maps of annual mean sea ice velocity (arrow, cm s$^{-1}$) from AOGCM experiments.
(a) MIS3 and (b) differences between MIS3 and MIS3-5aice. The results of the last 100 years are
used.

>L329: ice sheet"-induced" cooling?
Corrected.

>L335: replace "deny" with "exclude"?
Corrected.

>L348: "resemble"?
Corrected. We moved this paragraph to the supplement to increase the eligibility of the section.

>Fig. 9: the color of "PC-MIS3-5aice" in the legend is not correct.
We will fix the legend and also modify the color of "PC-MIS3-5aice" as follows.

[Figure]

Figure 10: Results of partially coupled experiment conducted with the AOGCM. (a) Time series of the maximum strength of the AMOC. (b) Spatial pattern of the Atlantic meridional streamfunction calculated from PC-MIS3heat. The climatology of the last 100 years is used to create this figure.

**New Reference**

Born, A., Nisancioglu, K. H., and Braconnot, P.: Sea ice induced changes in ocean circulation during the Eemian, Climate Dynamics, 35, 1361-1371, 10.1007/s00382-009-0709-2, 2010.

Hu, A. X., Meehl, G. A., Han, W. Q., Otto-Bliestner, B., Abe-Ouchi, A., and Rosenbloom, N.: Effects of the Bering Strait closure on AMOC and global climate under different background climates, Progress in Oceanography, 132, 174-196, 10.1016/j.pocean.2014.02.004, 2015.

Otto-Bliesner, B. L., Hewitt, C. D., Marchitto, T. M., Brady, E., Abe-Ouchi, A., Crucifix, M., Murakami, S., and Weber, S. L.: Last Glacial Maximum ocean thermohaline circulation: PMIP2 model intercomparisons and data constraints, Geophysical Research Letters, 34, 6, 10.1029/2007gl029475, 2007.

Rasmussen, S. O., Abbott, P. M., Blunier, T., Bourne, A. J., Brook, E., Buchardt, S. L., Buizert, C., Chappellaz, J., Clausen, H. B., Cook, E., Dahl-Jensen, D., Davies, S. M., Guillevic, M., Kipfstuhl, S., Laepple, T., Seierstad, I. K., Severinghaus, J. P., Steffensen, J. P., Stowasser, C., Svensson, A., Vallelonga, P., Vinther, B. M., Wilhelms, F., and Winstrup, M.: A first chronology for the North Greenland Eemian Ice Drilling (NEEM) ice core, Climate of the Past, 9, 2713-2730, 10.5194/cp-9-2713-2013, 2013.

Weber, S. L., Drijfhout, S. S., Abe-Ouchi, A., Crucifix, M., Eby, M., Ganopolski, A., Murakami, S., Otto-Bliesner, B., and Peltier, W. R.: The modern and glacial overturning circulation in the Atlantic Ocean in PMIP coupled model simulations, Climate of the Past, 3, 51-64, 2007.

---

## Author Comment (AC2) · 2 Sep 2020

Reply to Reviewer2

We are grateful to the reviewer for his/her time in evaluating the manuscript as well as for constructive comments and suggestions. As listed below, we have taken all the comments into account by the reviewer in the revised manuscript. In the following, our responses will be written in blue, while the comments by the reviewer will be written in black.

Major comments:
1.  In Figure 1, please add a panel for the ice-sheet anomalies between 36ka and 80ka, since it is a key to interpolate the modelling results.
We will add the following figure in the revised manuscript as Fig. 2.

[Figure]

Figure 2: Surface Topography of (a) MIS5a (80 ka), (b) MIS3 (36 ka), and their difference (c) MIS3 - MIS5a. Results from an ice sheet model are presented (Abe-Ouchi et al. 2013). These ice sheet configurations are used for climate model simulations.

2. Line 116: why to use the CO2 concentration and insolation at 35ka, instead of 36ka? A linguistic error? Or specific reason?
To be honest, there is no special reason. At the time we started the experiment, we had the data of the ice sheet of 36ka in our server, hence we used it. Nevertheless, there is very little difference in the simulated ice sheet between 36ka and 35ka. Therefore, we don't think this slight difference in the ice sheet will affect our result.

3. Line 170: Please add a reference for the LGM experiment.
We will add the reference of LGM experiment (Sherriff-Tadano and Abe-Ouchi 2020) in the revised manuscript.

4. Line 195-196: Please give the value of AMOC strength in PI experiment.
We will include the value (16.1 Sv) in the revised Table 1.
Table 1: Forcing and boundary conditions of climate simulations. Results of global mean temperature (GMT) and Atlantic meridional overturning circulation (AMOC) are also shown.

| Name | $CO_2$ | Ice sheet | Obliquity | Precession | Ecc | GMT | AMOC |
|---|---|---|---|---|---|---|---|
| MIS5a | 240 ppm | 80 ka | 23.175 | 312.25 | 0.0288 | 10.58°C | 18.7 Sv |
| MIS3 | 200 ppm | 36 ka | 22.754 | 251.28 | 0.0154 | 7.85°C | 15.6 Sv |
| MIS3-5aice | 200 ppm | 80 ka | 22.754 | 251.28 | 0.0154 | 8.91°C | 15.1 Sv |
| PI | 285 ppm | 0 ka | 23.45 | 102.04 | 0.0167 | 12.83°C | 16.1 Sv |

5. Line 219 and Figure 6: Please add the curve for the modelled PI state.
We will add the result of PI in the revised Fig. 7.

[Figure]

Figure 7: Northward oceanic heat transport over the Atlantic basin simulated from the AOGCM. Red: MIS3, Green: MIS3-5aice, and Black: PI. The climatology of the last 100 years is used to create these figures.

6. In what area are the NADW formed? Are they consistent among experiments? Any response of the NADW formation in the NORDIC Sea?
The deepwater mainly forms at the Nordic Sea and Irminger Sea. They are similar among the experiments, but there is a slight southward shift in the Nordic Seas in MIS3 compared with MIS5a. We will add a figure of deepwater formation region in the supplementary figure.

[Figure]

Figure S1: Spatial map of sea ice edge (contour) and deepwater formation region (color) at the North Atlantic. For sea ice, climatology of 15% sea ice concentration at February (solid) and August (dashed) are shown. For deepwater formation region, frequency of convective adjustment at 600 meter depth is shown. The climatology of the last 100 years is used to create these figures

7. Line 249: bottom ocean stratification with respective to density? If so, please add the information for density in Figure 4.

Yes, in deed. We will add a figure of density in the revised figure 5. Also, we noticed that the previous figure was showing the zonal average of the global ocean. We fixed this mistake in the revised manuscript.

[Figure]

Figure 5: Anomalies of zonally averaged oceanic properties over the Atlantic simulated from the AOGCM. The top panels show temperature anomalies, the middle panels show salinity anomalies, and the bottom panels show density anomalies. (a, d, g) MIS5a minus PI, (b, e, h) MIS3 minus PI, (c, f, i) MIS3 minus MIS3-5aice. The climatology of the last 100 years is used to create these figures.

8. Line 271: In addition to Figure 10, please show the convection map as that in Fig.7c.
We will add a figure of the convection map in the revised figure S1 as shown above.

9. Also in Line 271: please add a figure for the statement 'colder water occupies the subsurface ocean in MIS3 compared with MIS3-5aice.', in either Main text or SI.
We will add a figure of the vertical profile of ocean temperature in the supplementary figure.

[Figure]

Figure S2: Vertical profile of oceanic properties at the North Atlantic Deep Water formation region (60˚W-0˚, 55˚N-65˚N). Red: MIS3 and Green: MIS3-5aice. Cold water occupies the subsurface ocean in MIS3 compared with MIS3-5aice. The climatology of the last 100 years is used to create these figures.

10. In Table 1: please add the information also for the PI and LGM experiments together with their references.

We will add the information of PI in the revised Table 1 since the results of PI appear in several figures (please see the revised Table 1 shown above). For LGM, we decided not to add in the table, as the results do not appear in other figures, and it is discussed only for once. Nevertheless, we will add a reference of LGM in the revised manuscript.

11. In Figure 11, how to address the impact of stronger surface winds on the northward ocean heat transport and surface cooling in the northern North Atlantic? Any indications based on the experiments in this study?

As pointed out by the reviewer, the stronger wind can increase the strength of the northward oceanic heat transport at high latitude by strengthening the wind-driven ocean circulation. This can then increase the strength of the surface cooling (or atmosphere-ocean heat exchange) by increasing the temperature difference between the atmosphere and ocean. This result implies that the strengthening of the surface wind by the ice sheets also increases the strength of the surface cooling, and hence may work as a brake of the AMOC strengthening. We will add a following paragraph on this point in the revised manuscript:

"While the present study shows the individual effect of surface cooling and surface wind on the AMOC by means of partially coupled experiment, we should note that these two components can interact. For example, the strengthening of the surface wind can increase the strength of the northward oceanic heat transport at high latitude by strengthening the wind-driven ocean circulation. This can then increase the strength of the surface cooling (or atmosphere-ocean heat exchange) by increasing the temperature difference between the atmosphere and ocean. This result implies that the strengthening of the surface wind also increases the strength of the surface cooling, and hence may work as a brake of the AMOC strengthening. Further studies will be required to assess the interaction of these two effects."

Also, we decided to remove the information of internal feedback in the schematic figure to make the figure simple and to focus on the main topic of this paper, which is to show that the impact of the ice sheet is determined by the two competing effects, surface wind and surface cooling. Nevertheless, we will keep the discussion on the internal feedback in the revised manuscript.

Minor comments:
Line 37: 'Project' to 'Projects'
Corrected.

Line 210: please refer to Figure 2d, for the warmer surface around Alaska
Corrected.

Line 303: '.' has been double used.
Corrected. Thank you for pointing out.

---

## Author Response (AR1)

Reply to Editor

We are grateful to the editor for the time in evaluating our manuscript and for constructive comments and suggestions, which have helped to improve the quality of our manuscript. As listed below, we have taken all the comments into account in the revised manuscript. In the following, our responses will be written in blue and the comments by the editor will be written in black.

L. 329-330: The meaning of that sentence is unclear to me.
Following the comment, we modified the corresponding sentence in the revised manuscript as follows (L347-349),
"For example, if glacial ice sheets cause a very small cooling over the Southern Ocean in other models, this will reduce the weakening effect of the AMOC through the Southern Ocean (Fig. 12). As a result, the overall weakening effect by the glacial ice sheet induced-cooling on the AMOC should be reduced."

L. 337 (and elsewhere): Please consider changing "strengthening of surface cooling" into "a stronger surface cooling" or "a decrease in surface temperature".
We changed it to " a stronger surface cooling" (L357).
This modification is also applied in L258, L342, L419.

Please remove the bullet points from the Conclusion
Following the editor's suggestion, we removed the bullet points.

L. 419-420: Please slightly amend the sentence so that it is clear that some climate model outputs are available for MIS3 (even if less so than for the LGM), and consider also adding the MIS3 simulation performed with LOVECLIM.
Thank you for pointing out. Indeed, we should include results from LOVECLIM, which also offer very useful dataset of MIS3. We modified the sentence as follows (L431-433),
"The results of the present study also offer a global dataset of climate during MIS3 and MIS5a, which has been also provide by previous studies (e.g. Van Meerbeeck et al. 2009, Gong et al. 2013, Menviel et al. 2014, Guo et al. 2019), though still lacking compared with the LGM"

Figure 9a: Consider changing the colours of some lines (e.g. the 2 green and 2 purple lines can be confusing).
We modified the colors in the revised figure 10a. We hope the revised figure is easier to see.

Schematic (Fig. 11): The impact of Southern Ocean sea-ice on AMOC is unclear to me
We added a following sentence in the caption of revised figure 12 to make clearer the effect of Southern Ocean sea-ice on AMOC.
"A stronger surface wind induced by the glacial ice sheets enhances wind-driven transport of salt into the deepwater formation region and causes a strengthening of the AMOC. In contrast, a stronger surface cooling by the glacial ice sheets causes a weakening of the AMOC by increasing the sea ice at the North Atlantic, which insulates the atmosphere-ocean heat exchange (Oka et al. 2012). A stronger surface cooling by the northern glacial ice sheets also causes a cooling and an increase in sea ice over the Southern Ocean by increasing the oceanic heat transport. This change in the Southern Ocean then weakens the AMOC by increasing the density of the AABW and bottom ocean stratification (Weber et al. 2007, Klockmann et al. 2018)."

Answer to Reviewer 2 (point 11): You state that stronger wind, strengthen the wind driven circulation and meridional oceanic heat transport (as seen in Fig. 6). However, you also suggest that this leads to further cooling in the North Atlantic. I agree that stronger wind will enhance ocean heat release in the North Atlantic, acting to cool the ocean and warm the atmosphere. However the stronger wind-driven circulation and enhanced meridional oceanic heat transport should lead to a warming of the North Atlantic.
Thank you for the constructive comment. We agree to the editor's point. In the reply to Reviewer 2, we mainly considered the changes in the atmosphere-ocean heat flux. However, as mentioned by the editor, the increase in the oceanic heat transport by the surface wind does increase the surface temperature at high latitude. Based on a previous study (Oka et al. 2012), showing the importance of surface temperature on the glacial AMOC through its effect on the sea ice, we reconsider that the changes in the surface temperature is more important than the atmosphere-ocean heat flux itself. Following this reconsideration, we modified the corresponding paragraph as follows in L316-322.

"Considering the fact that most climate models show a strengthening of the AMOC in response to the glacial ice sheet expansion, the effect of surface wind seems to dominate in most models. The reason behind this still remains elusive, though we speculate that two processes play a role. The first process is associated with the change in wind-driven transport of heat over the subpolar region. For example, the strengthening of the surface wind can increase the strength of the northward oceanic heat transport at high latitude by enhancing the wind-driven ocean circulation. This causes an increase in the surface air temperature and a decrease in sea ice at high latitudes and can reduce the effect of a stronger surface cooling by the glacial ice sheets. "

Figure S1 (In the answer to Reviewer 2), c) should be MIS3-5aice
Thank you for pointing out. We corrected this mistake.

Reply to Reviewer1

We are grateful to the reviewer for his time in evaluating the manuscript and his constructive comments and suggestions, which have helped to improve the quality of our manuscript. As listed below, we have taken all the comments into account by in the revised manuscript. In the following, our responses will be written in blue, and the comments by the reviewer will be written in black.

>L24-34: I suggest that the authors add a schematic figure illustrating the time evolution of certain climate variables from paleo records (e.g. summer insolation, CO2, sea level, d18O etc.) from last interglacial to the present day. This can provide a more clear context and would be especially beneficial to a wider audience.

We agree to the suggestion. We add a figure illustrating the time evolution of summer insolation, CO2, sea level, Greenland ice core data, and AMOC in Fig.1 of the revised manuscript.

[Figure]

Figure 1: Time series of climate records of the last glacial period. (a) 65˚N July insolation (W m$^{-2}$), (b) black: sea level data from Spratt and Lisiecki (2016), brown: sea level data from Grant et al. (2012), gray: simulated time evolution of ice sheet by Abe-Ouchi et al. 2013, (c) CO$_2$ (Bereiter et al. 2015), (d) Greenland ice core delta 18 O from North Greenland Ice Core Project (NGRIP) core (Rasumussen et al. 2013). (e) Bermuda Rise $^{231}$Pa/$^{230}$Th (Bohm et al. 2015), which is a proxy of the strength of the AMOC. Red and Blue shades correspond to the target period of MIS3 and MIS5a in our climate model simulations, respectively.

>section 2.1: Could the authors add a short paragraph briefly summarizing the performance of MIROC4m for the preindustrial and/or present day simulations, especially for the metrics that are relevant for the analysis later in the main text? Such metrics can include, but not limited to, sea ice concentration, mixed layer depth, ocean profiles/stratification in the North Atlantic. Climate sensitivity would be useful to mention as well. Any significant bias and therefore its implication for the conclusions drawn in this work should also be discussed where relevant.

Following the reviewer's suggestion, we add a following paragraph in the method section in the revised manuscript (L111-L118).

"The model version used in this study reproduces the modern AMOC (Fig. 6d), the deepwater formation over the Nordic Seas (Fig. S1) and sea ice extent over the North Atlantic (Fig. 4) reasonably well as in the previous version (Otto-Bliesner et al. 2007, Weber et al. 2007, Kawamura et al. 2017). While the current model version overestimates sea ice extent and lacks deepwater formation over the Labrador Sea (Fig. S1, Fig. 4), the performance of the modern Southern Ocean sea ice extent has improved compared with the previous version (Fig. 4). This model version has been used extensively for paleoclimate (Obase and Abe-Ouchi 2019) and future climate studies (Yamamoto et al. 2015). It has a climate sensitivity of 4.1 K and reproduces the AMOC of the LGM reasonably well (Sherriff-Tadano and Abe-Ouchi 2020)."

>L121: Did the authors perform any sensitivity experiment with regard to the opening/closing of the Bering Strait by any chance? If yes would be useful to briefly discuss it here. Some studies have shown how an opened/closed Bering Strait could have some significant impact on the North Atlantic ocean state.

As pointed out by the reviewer, the closure of the Bering Strait can have an impact on the AMOC. However, unfortunately, we have not performed sensitivity experiments closing the Bering Strait. Nevertheless, we add a following sentence to the revised manuscript so that the readers can refer to the effect of Bering Strait on the AMOC in other studies (L127-L129).

"The global sea level is unchanged, and the land sea mask outside the northern glacial ice sheet region is same as the modern configuration (e.g., the Bering Strait remains open, which itself may impact on the AMOC (Hu et al. 2015))."

>L355-357: It is not immediately clear to me how do subsurface warming and southern ocean warming are able to re-strengthen the AMOC. The latter due to reduced production of AABW? How about subsurface warming? Please elaborate a bit more on the dynamic links here.

We removed the sentence associated with the subsurface warming since we agree that the accumulation of heat at the subsurface ocean over the North Atlantic does not cause a gradual recovery of the AMOC, but rather causes an abrupt strengthening by triggering a new deepwater formation. With respect to the Southern Ocean process, we increase the explanation and also add some references, which show the effect of temperature changes over the Southern Ocean on the AMOC (Buizert and Schmittner 2015, Jansen 2017), to support our discussion. We also nuanced the paragraph since the original sentence looked too confident (L362-L365).

"With respect to the oceanic feedback, the weakening of the AMOC causes a warming over the Southern Ocean due to the reduction in the northward heat transport, and hence reduces the deep ocean stratification and the density of AABW. These processes may contribute to re-strengthen the AMOC (Weber et al. 2007, Buizert and Schmittner 2015, Jansen 2017, Klockmann et al. 2018). "

>L358-359: Once again, it is not clear to me the link between the expanded sea ice and a weakening surface wind. My understanding is that a more extensive sea ice cover in the North Atlantic 'protects' the ocean surface from the wind stress above, which tends to spin down the ocean circulation, and is favorable for maintaining a weak AMOC.

As the reviewer says, the link between the expanded sea ice and a weakening of the surface wind was not clear in the original manuscript. In the revised manuscript, we increased the description on this topic as follows (L365-L372);

"With respect to the lack of atmospheric feedback, previous studies show that the expansion of sea ice due to a reduction of the AMOC causes a weakening of the surface wind over the North Atlantic by increasing the static stability of the lower troposphere (Byrkedal et al. 2006, Sherriff-Tadano and Abe-Ouchi 2020). They further show that this weakening of the surface wind plays a role in maintaining a weak AMOC by reducing the wind-driven transport of salt to the deepwater formation region (Zhang et al. 2014a, Sherriff-Tadano and Abe-Ouchi 2020). In the partially coupled experiments described above (PC-MIS3heat), the sea ice covers the large area of northern high latitude (Fig. S1e), and should activate this positive feedback, which will stabilize the weak AMOC. However, these atmospheric feedbacks are removed in PC-MIS3heat and hence may contributed to the destabilization of the weak AMOC."

Also, as the reviewer pointed out, the extensive sea ice protects the ocean surface from the wind stress, and causes a weakening of the wind-driven ocean circulation. In fact, this feedback is taken into account in the model experiments (both the original and partially coupled) when the sea ice expands. However, if this feedback is dominant, one should expect a stable weak AMOC, rather than a gradual increase in the AMOC, which is observed in our partially coupled experiments (Fig. 10a). This result suggests that other processes/feedback after the weakening of the AMOC is causing the gradual increase in the AMOC. From several previous studies presented above, we speculate that the lack of the sea ice-wind feedback can destabilize the weak AMOC and cause the gradual increase in the AMOC.

>L340-364: I appreciate the authors' efforts in explaining some of the interesting modelling results here. However, to me this part has a very limited contribution to the main points of the paper, and could be a distraction to the readers in this section. I think by removing it or moving it to supplementary material could help enhance the legibility of this section. It is up to the authors to decide though.

Following the reviewer's suggestion, we decided to move the first paragraph to the supplement to increase the eligibility of the section. Related to this, we moved Fig. 10c,d in the original manuscript to Fig. S3 in the revised manuscript because it is no longer discussed in the main manuscript. For the second paragraph (L360-L375 in the revised manuscript), we decided to keep it in the main manuscript since we think this paragraph discusses important internal feedback within the atmosphere-ocean system, which can appear in the simple schematic figure 12.

>Fig. 11: this schematic is not adequately discussed/referred to in the main text. There are several places in the text (mainly in 'Discussion') where the relevant processes are described and should refer to this figure. In addition, the feedbacks indicated by the black solid arrows are not straightforward to me. Please consider elucidating it more explicitly in the main text or in the caption where appropriate.

In the revised manuscript, we now adequately refer to the schematic figure (Fig. 12 in the revised manuscript) in 4.2, 5, and 6. We also increased the explanation of each effect in the caption so that the reader can understand the figure more easily. In addition, in order to concentrate on our main

finding, we decided to remove the black arrows of internal feedback from the revised Fig. 12. Nevertheless, we modified the caption so that the reader can refer to the main text for the discussion on the possible internal feedbacks, which is increased as described above.

[Figure]

Figure 12: Simple schematic of the processes by which changes in the glacial ice sheet affect the AMOC. A stronger surface wind induced by the glacial ice sheets enhances wind-driven transport of salt into the deepwater formation region and causes a strengthening of the AMOC. In contrast, a stronger surface cooling by the glacial ice sheets causes a weakening of the AMOC by increasing the sea ice at the North Atlantic, which insulates the atmosphere-ocean heat exchange (Oka et al. 2012). A stronger surface cooling by the northern glacial ice sheets also causes a cooling and an increase in sea ice over the Southern Ocean by increasing the oceanic heat transport. This change in the Southern Ocean then weakens the AMOC by increasing the density of the AABW and bottom ocean stratification (Weber et al. 2007, Klockmann et al. 2018). Possible internal feedbacks within the atmosphere-sea ice-ocean system are discussed in the Discussion section.

Minor and technical comments:
>title: I think that it is good practice to try to avoid abbreviations in the title (e.g. AMOC).
Corrected.

>L8: should spell out that it is about the expansion of ice sheet in North America.
Corrected (L9).

>L10: it would be useful to mention the MIS3 and 5a time slices that the authors chose in this study, such that the readers can get a quick grasp by reading the abstract.
We add the time slice in the revised manuscript (L11) as well as in the revised Fig. 1.

>L55-59: suggest to rephrase the sentence as ": : :, which can cause either a strengthening of the AMOC by : : :, or a weakening of the AMOC by: : :" This also applies to L246-248.
Corrected (L57-L60 and L259).

>L65: you mean "For" these two periods?
We meant that the ice sheet of MIS3 is considered to be slightly larger compared with that of MI5a (L67).

>L71: ": : :, whose effect of surface cooling is prominent." This reads a bit ambiguous to me; please consider rephrasing it.
We modified this sentence in the revised manuscript as follows (L72-L73);
"Hence, by comparing the early-glacial and mid-glacial ice sheets, one may obtain different responses in the AMOC and global climate and quantify the effect of changes in  ice sheet extent and the surface cooling"

>L108-109: is it relevant to include the information in the square bracket? If not please consider removing it.
We think this sentence is important to avoid readers from getting confused when they look back to previous articles. Therefore, we will remain this sentence (L110-L111).

>L146-147: To my understanding, it should be stressed that surface heat flux cannot be imposed because it is strongly coupled to SST, whereas surface freshwater flux can because there is no direct SSS feedback to the flux.
As suggested by the reviewer, we modified the sentence as follows (L153-L156),
"Following previous studies (Schmittner et al. 2002, Gregory et al. 2005), the heat flux is unchanged in these experiments. This is because the heat flux is strongly coupled to the sea surface temperature and that fixing the surface heat condition has an unrealistic impact on the AMOC (Marozke 2012)."

>L168-170: I am a bit surprised the simulated LGM climate is only about 0.2 deg-C colder than the MIS3 climate, considering that there is a $CO_2$ difference of 20 ppm plus some (supposedly moderate) difference in the distribution of ice sheet. Could the authors comment on this?
We were also surprised at the result when we saw it. One possible reason is the lower obliquity in this experiment compared with the LGM. Several previous studies have shown that the lower obliquity can cause an annual global cooling by increasing the sea ice in the Southern Ocean and Arctic regions, even though the global input of insolation does not change. We added a following sentence on this point in the revised manuscript (L179-L181).
"The strong MIS3 cooling similar to that of LGM is possibly related to the low obliquity applied in MIS3, which increases the amount of sea ice in both hemispheres and causes a global cooling through feedbacks within the atmosphere-ocean coupled system (Galbraith and de Lavergne 2019). "

>L184: perhaps the reference of Dokken et al. and Sadazki et al. in lines 193-194 can be moved here.
We added these reference in the sentence (L195).

>L186: I find it a bit odd to say "the western part of the Southern Ocean"; suggest to change to, for example, Pacific/Indian/Atlantic Ocean sector of the Southern Ocean.

Corrected (L197).

>L220: it is not clear from Fig. 4 that there is 'stronger surface cooling'. I see a relatively homogenous distribution of ocean cooling in Fig 4(a,b). Is this the case or it has to do with the color bar?

As pointed out by the reviewer, the original sentence was misleading. We should have clearly mention that we are referring to Fig. 5c in the revised manuscript, which shows the effect of ice sheet on ocean temperature. We have modified this sentence as follows (L231-L232),

"This is associated with a cooling of the NADW (Fig. 5c), which is induced by the stronger surface cooling by the glacial ice sheets. "

In Fig.5c, you can find a cooling of NADW, which is associated with the ice sheet expansion and the resulting stronger surface cooling.

>L222: "and increases the deep ocean salinity, : : :" error in grammar. Also, should spell out the increased deep ocean salinity is via brine rejection.

Corrected. Also added the explanation of brine rejection (L233-L235).

>L235: suggest to move "Fig. 7c,d" to the middle of L234.

Corrected (L246).

>L261: change "are replaced with" to "replace with"?

As pointed out by the reviewer, this sentence was strange. We modified the sentence as follows (L273-L274).

"In the third experiment (PC-MIS3heat), in which the monthly climatology of surface wind stress and atmospheric freshwater flux of MIS3 are replaced with those of MIS3-5aice (Table 2)"

>L265: "compensates"

Corrected (L278).

>L269-271: "Due to : : : AMOC (Fig. 10b)." To me the main effect of sea ice in weakening the AMOC in the north Atlantic is because of its insulation that reduces air-sea flux and therefore ocean convection. The effect of melting of sea ice, if one can do a back-of-envelope calculation converting the melted sea ice into sverdrups, should be relatively small.

As the reviewer suggests, the expansion of sea ice weakens the AMOC by suppressing the atmosphere-ocean heat exchange (Oka et al. 2012). In addition, it has been shown that the increase in sea ice over the north North Atlantic can reduce the AMOC and the ocean convection via meltwater at the sea ice edge (Born et al. 2010). Following these previous studies, we modify this sentence as follows (L282-L285);

"Due to this surface cooling, the sea ice increases over the northern North Atlantic (Fig. 11b). The increase in sea ice tends to weaken the oceanic convection and the AMOC by insulating the atmosphere-ocean heat flux (Oka et al. 2012) and by increasing the meltwater flux over the deep-water formation region (Born et al. 2010). "

>L272: again, the more stable ocean column is not clear to me from Fig. 4c.

We add a figure of vertical profile of ocean temperature in the Fig. S2, which shows that MIS3 exhibits more stable ocean column in terms of temperature compared with MIS3-5aice.

[Figure]

Figure S2: Vertical profile of oceanic properties at the North Atlantic Deep Water formation region (60˚W-0˚, 55˚N-65˚N). Red: MIS3 and Green: MIS3-5aice. Cold water occupies the subsurface ocean in MIS3 compared with MIS3-5aice. The climatology of the last 100 years is used to create these figures.

>L273: suggest to tone down "overcomes" to "tends to overcome".
Corrected (L287).

>L283: "The results above demonstrate: : :"?
Corrected (L297).

>L303: there are two full stops.
Corrected (L317). Thanks for pointing out.

>L303-307: this reads very speculative to me, if I understand the authors' point correctly here. Please consider removing it or providing more evidence (it's up to the authors to decide).
Indeed it is a speculative discussion, but we think this point is quite important, which the modelers should keep in mind. We add a figure supporting this sentence in the supplementary file and keep this discussion in the revised manuscript (L324).

[Figure]

Figure S4: Spatial maps of annual mean sea ice velocity (arrow, cm s⁻¹) from AOGCM experiments. (a) MIS3 and (b) differences between MIS3 and MIS3-5aice. The results of the last 100 years are used.

>L329: ice sheet"-induced" cooling?
Corrected (L349).

>L335: replace "deny" with "exclude"?
Corrected (L356).

>L348: "resemble"?
Corrected. We moved this paragraph to the supplement to increase the eligibility of the section.

>Fig. 9: the color of "PC-MIS3-5aice" in the legend is not correct.
We fix the legend and also modify the color of "PC-MIS3-5aice" as follows. Note that the figure has been modified slightly from that submitted in the previous reply letter following editor's suggestion.

[Figure]

[Figure]

Figure 10: Results of partially coupled experiment conducted with the AOGCM. (a) Time series of the maximum strength of the AMOC. (b) Spatial pattern of the Atlantic meridional streamfunction calculated from PC-MIS3heat. The climatology of the last 100 years is used to create this figure.

**New Reference**

Born, A., Nisancioglu, K. H., and Braconnot, P.: Sea ice induced changes in ocean circulation during the Eemian, Climate Dynamics, 35, 1361-1371, 10.1007/s00382-009-0709-2, 2010.

Hu, A. X., Meehl, G. A., Han, W. Q., Otto-Bliestner, B., Abe-Ouchi, A., and Rosenbloom, N.: Effects of the Bering Strait closure on AMOC and global climate under different background climates, Progress in Oceanography, 132, 174-196, 10.1016/j.pocean.2014.02.004, 2015.

Otto-Bliesner, B. L., Hewitt, C. D., Marchitto, T. M., Brady, E., Abe-Ouchi, A., Crucifix, M., Murakami, S., and Weber, S. L.: Last Glacial Maximum ocean thermohaline circulation: PMIP2 model intercomparisons and data constraints, Geophysical Research Letters, 34, 6, 10.1029/2007gl029475, 2007.

Rasmussen, S. O., Abbott, P. M., Blunier, T., Bourne, A. J., Brook, E., Buchardt, S. L., Buizert, C., Chappellaz, J., Clausen, H. B., Cook, E., Dahl-Jensen, D., Davies, S. M., Guillevic, M., Kipfstuhl, S., Laepple, T., Seierstad, I. K., Severinghaus, J. P., Steffensen, J. P., Stowasser, C., Svensson, A., Vallelonga, P., Vinther, B. M., Wilhelms, F., and Winstrup, M.: A first chronology for the North Greenland Eemian Ice Drilling (NEEM) ice core, Climate of the Past, 9, 2713-2730, 10.5194/cp-9-2713-2013, 2013.

Weber, S. L., Drijfhout, S. S., Abe-Ouchi, A., Crucifix, M., Eby, M., Ganopolski, A., Murakami, S., Otto-Bliesner, B., and Peltier, W. R.: The modern and glacial overturning circulation in the Atlantic Ocean in PMIP coupled model simulations, Climate of the Past, 3, 51-64, 2007.

Reply to Reviewer2

We are grateful to the reviewer for the time in evaluating our manuscript and for constructive comments and suggestions, which have helped to improve the quality of our manuscript. As listed below, we have taken all the comments into account in the revised manuscript. In the following, our responses will be written in blue, and the comments by the reviewer will be written in black.

Major comments:
1. In Figure 1, please add a panel for the ice-sheet anomalies between 36ka and 80ka, since it is a key to interpolate the modelling results.
We agree on this point. We add the following figure in the revised manuscript as Fig. 2.

[Figure]

Figure 2: Surface Topography of (a) MIS5a (80 ka), (b) MIS3 (36 ka), and their difference (c) MIS3 - MIS5a. Results from an ice sheet model are presented (Abe-Ouchi et al. 2013). These ice sheet configurations are used for climate model simulations.

2. Line 116: why to use the CO2 concentration and insolation at 35ka, instead of 36ka? A linguistic error? Or specific reason?
To be honest, there is no special reason. At the time we started the experiment, we had the data of the ice sheet of 36ka in our server, hence we used it. Nevertheless, there is very little difference in the simulated ice sheet between 36ka and 35ka. Therefore, we don't think this slight difference in the ice sheet will affect our result.

3. Line 170: Please add a reference for the LGM experiment.
We add the reference of LGM experiment (Sherriff-Tadano and Abe-Ouchi 2020) in the revised manuscript (L179).

4. Line 195-196: Please give the value of AMOC strength in PI experiment.
We include the value (16.1 Sv) in the revised Table 1.
Table 1: Forcing and boundary conditions of climate simulations. Results of global mean temperature (GMT) and Atlantic meridional overturning circulation (AMOC) are also shown.

| Name | CO₂ | Ice sheet | Obliquity | Precession | Ecc | GMT | AMOC |
|---|---|---|---|---|---|---|---|
| MIS5a | 240 ppm | 80 ka | 23.175 | 312.25 | 0.0288 | 10.58°C | 18.7 Sv |
| MIS3 | 200 ppm | 36 ka | 22.754 | 251.28 | 0.0154 | 7.85°C | 15.6 Sv |
| MIS3-5aice | 200 ppm | 80 ka | 22.754 | 251.28 | 0.0154 | 8.91°C | 15.1 Sv |
| PI | 285 ppm | 0 ka | 23.45 | 102.04 | 0.0167 | 12.83°C | 16.1 Sv |

5. Line 219 and Figure 6: Please add the curve for the modelled PI state.
We add the result of PI in the revised Fig. 7.

[Figure]

Figure 7: Northward oceanic heat transport over the Atlantic basin simulated from the AOGCM. Red: MIS3, Green: MIS3-5aice, and Black: PI. The climatology of the last 100 years is used to create these figures.

6. In what area are the NADW formed? Are they consistent among experiments? Any response of the NADW formation in the NORDIC Sea?
The deepwater mainly forms at the Nordic Sea and Irminger Sea. They are similar among the experiments, but there is a slight southward shift in the convection site at the Nordic Seas in MIS3 compared with MIS5a.
We add a figure of deepwater formation region in the supplementary figure.

[Figure]

Figure S1: Spatial maps of sea ice edge (contour) and deepwater formation region (color) at the North Atlantic. For sea ice, climatology of 15% sea ice concentration at February (solid) and August (dashed) are shown. For deepwater formation region, frequency of convective adjustment at 600 meter depth is shown. The climatology of the last 100 years is used to create these figures

7. Line 249: bottom ocean stratification with respective to density? If so, please add the information for density in Figure 4.
Yes, in deed. We add a figure of density in the revised figure 5. Also, we noticed that the previous figure was showing the zonal average of the global ocean. We fixed this mistake in the revised manuscript.

[Figure]

Figure 5: Anomalies of zonally averaged oceanic properties over the Atlantic simulated from the AOGCM. The top panels show temperature anomalies, the middle panels show salinity anomalies, and the bottom panels show density anomalies. (a, d, g) MIS5a minus PI, (b, e, h) MIS3 minus PI, (c, f, i) MIS3 minus MIS3-5aice. The climatology of the last 100 years is used to create these figures.

8. Line 271: In addition to Figure 10, please show the convection map as that in Fig.7c.
We add a figure of the convection map in the revised figure S1 as shown above.

9. Also in Line 271: please add a figure for the statement 'colder water occupies the subsurface ocean in MIS3 compared with MIS3-5aice.', in either Main text or SI.
We add a figure of the vertical profile of ocean temperature in the supplementary figure.

[Figure]

Figure S2: Vertical profile of oceanic properties at the North Atlantic Deep Water formation region (60˚W-0˚, 55˚N-65˚N). Red: MIS3 and Green: MIS3-5aice. Cold water occupies the subsurface ocean in MIS3 compared with MIS3-5aice. The climatology of the last 100 years is used to create these figures.

10. In Table 1: please add the information also for the PI and LGM experiments together with their references.
We add the information of PI in the revised Table 1 since the results of PI appear in several figures (please see the revised Table 1 shown above). For LGM, we decided not to add in the table, as the results do not appear in other figures, and it is discussed only for once. Nevertheless, we add a reference of LGM in the revised manuscript (L179).

11. In Figure 11, how to address the impact of stronger surface winds on the northward ocean heat transport and surface cooling in the northern North Atlantic? Any indications based on the experiments in this study?
Thank you for the constructive comment. We reconsidered our previous response to Reviewer 2 after receiving editor's comment. In the previous reply, we mainly considered the changes in the atmosphere-ocean heat flux associated with the surface wind anomaly. In this case, the stronger oceanic wind-driven heat transport will increase the temperature difference between the atmosphere and ocean, and causes an increase in the atmosphere-ocean heat flux. However, as mentioned by the editor, the increase in the oceanic heat transport induced by the surface wind anomaly does try to increase the surface temperature at high latitude. Based on a previous study (Oka et al. 2012), showing the importance of surface temperature on the glacial AMOC through its effect on the sea ice, we reconsider that the changes in the surface temperature is more important than the atmosphere-ocean heat flux itself. Following this reconsideration, we reconsidered that the strong surface wind tries to reduce the weakening effect of the stronger surface cooling on the AMOC by increasing the surface temperature and reducing the sea ice, which is the opposite to what we have mentioned in the previous reply. The modified paragraph can be found in L316-322.
"Considering the fact that most climate models show a strengthening of the AMOC in response to the glacial ice sheet expansion, the effect of surface wind seems to dominate in most models. The reason behind this still remains elusive, though we speculate that two processes play a role. The first process is associated with the change in wind-driven transport of heat over the subpolar region. For example, the strengthening of the surface wind can increase the strength of the northward oceanic heat transport at high latitude by enhancing the wind-driven ocean circulation. This causes an increase in the surface air temperature and a decrease in sea ice at high latitudes and can reduce the effect of a stronger surface cooling by the glacial ice sheets. "

Also, we decided to remove the information of internal feedback in the schematic figure to make the figure simple and to focus on the main topic of this paper, which is to show that the impact of the ice sheet is determined by the two competing effects, surface wind and surface cooling. Nevertheless, we keep the discussion on the internal feedback in the revised manuscript.

Minor comments:
Line 37: 'Project' to 'Projects'
Corrected (L39).

Line 210: please refer to Figure 2d, for the warmer surface around Alaska
Corrected (L222).

Line 303: '.' has been double used.
Corrected (L317). Thank you for pointing out.

[revised manuscript text omitted]

**(a) Time series of maximum AMOC**

| | |
|---|---|
| MIS3 | |
| MIS3-5aice | |
| PC-MIS3 | |
| PC-MIS3-5aice | |
| PC-MIS3heat | |
| PC-MIS3heatano | |
| PC-MIS3day | |

[Figure]

**(b) Spatial pattern of AMOC for PC-MIS3heat**

[Figure]

**Figure 9̶10: Results of partially coupled experiment conducted with the AOGCM. (a) Time series of the maximum strength of the**
AMOC. **(b) Spatial pattern of the Atlantic meridional streamfunction calculated from PC-MIS3heat. The climatology of the last 100**
**years is used to create this figure.**

765

[Figure]

**Figure 1011: Annual mean sea ice thickness (cm, colour) over the North Atlantic simulated from the AOGCM and partially coupled experiments. (a) MIS3. (b) Effect of surface cooling by mid-glacial ice sheet (PC-MIS3heat minus PC-MIS3-5aice). (c) and (d) show**

the reproducibility of sea ice thickness by the partially coupled experiment: (c) PC-MIS3 minus MIS3 and (d) PC-MIS3day minus MIS3. In (a), (b), and (c), the results of the last 100 years are used. In (d), the results of the last 50 years are used for PC-MIS3day.The results of the last 100 years are used.

[Figure]

775

[Figure]

Figure 12: Simple schematic of the processes by which changes in the glacial ice sheet affect the AMOC. A stronger surface wind induced by the glacial ice sheets enhances wind-driven transport of salt into the deepwater formation region and causes a
780    strengthening of the AMOC. In contrast, a stronger surface cooling by the glacial ice sheets causes a weakening of the AMOC by increasing the sea ice at the North Atlantic, which insulates the atmosphere-ocean heat exchange (Oka et al. 2012). A stronger surface cooling by the northern glacial ice sheets also causes a cooling and an increase in sea ice over the Southern Ocean by increasing the oceanic heat transport. This change in the Southern Ocean then weakens the AMOC by increasing the density of the AABW and bottom ocean stratification (Weber et al. 2007, Klockmann et al. 2018). Possible internal feedbacks within the atmosphere-sea ice-
785    ocean system are discussed in the Discussion section.

---

## Author Response (AR2)

Reply to editor (Dr. Menviel)
We are grateful to Dr. Menviel for having interests in our paper. We have modified the manuscript following the editor's minor comment. We have also uploaded our model result on https://ccsr.aori.u-tokyo.ac.jp/~tadano/, which was pointed out by the editor in the first submission. Lastly, thank you very much for spending your time in evaluating our manuscript.

Reply to Rev.1 (Dr. Guo)
We are grateful to Dr. Guo for his thorough and constructive comments, which helped to improve the quality of the manuscript. As listed below, we have taken all the comments into account in the revised manuscript. Thank you very much for spending your time in evaluating our manuscript.

> It is up to the authors to decide, but I wonder if it would make sense to also address the role of 'surface wind' in the title? I understand that the authors have a main focus on the cooling effect, but dynamically these two effects are competing with each other, and the authors have also repeatedly described these two effects in the abstract.

We decided to remove the subtitle following the editor's suggestion to make the tile simple.

> Figure 1b: should spell out that the grey line shows the sea level equivalent values.
Corrected.

> Figure 1e: please consider adding a legend/symbol to indicate that large/small Pa/Th values correspond to strong/weak AMOC.
Corrected.

> L10, L40: should both be 'Atlantic Meridional Overturning Circulation' – to be consistent with that in the title.
Corrected.

> L31: summer insolation and concentration of CO2 were relatively large during MIS3? Should be the opposite.
We modified the sentence as follow to clarify we were comparing MIS3 with MIS4.
"Then, the glacial ice sheets shrank during the mid-glacial period (MIS3), when the summer insolation and the concentration of $CO_2$ were relatively large compared to MIS4 "

> L75: should refer to Figure 1 before the reference.
Corrected.

> L256: The increase of SSS due to the changes in E-P, according to Figure 9, does not occur in the deep water formation region in this model; should probably mention this.
In Fig. 8, red color shows a decrease in precipitation (or E minus P). Hence, the E minus P *does* increase in the deep water formation region, especially over the Irminger Sea. Therefore the southward expansion of ice sheet can cause an increase in SSS over the deepwater formation region by affecting the E-P.

> L275: should stress that it is the "winter" sea ice covers…
Corrected.

> L446: "provided"
We modified the word "provided" to "compliment" following the editor's suggestion.

> L420-444: it seems the authors removed the bullet points; should then the format be adjusted accordingly? The editor can probably provide some instructions on this.
We reorganized this part following the editor's suggestion.